# ER-associated ubiquitin ligase HRD1 programs liver metabolism by targeting multiple metabolic enzymes

Juncheng Wei[1], Yanzhi Yuan[2], Lu Chen[3], Yuanming Xu[1], Yuehui Zhang[2], Yajun Wang[1], Yanjie Yang[4], Clara Bien Peek[5], Lauren Diebold[6], Yi Yang[1], Beixue Gao[1], Chaozhi Jin[2], Johanna Melo-Cardenas[1], Navdeep S. Chandel[6], Donna D. Zhang[7], Hui Pan[3], Kezhong Zhang[8], Jian Wang[2], Fuchu He[2] & Deyu Fang[1]

The HMG-CoA reductase degradation protein 1 (HRD1) has been identified as a key enzyme for endoplasmic reticulum-associated degradation of misfolded proteins, but its organ-specific physiological functions remain largely undefined. Here we show that mice with HRD1 deletion specifically in the liver display increased energy expenditure and are resistant to HFD-induced obesity and liver steatosis and insulin resistance. Proteomic analysis identifies a HRD1 interactome, a large portion of which includes metabolic regulators. Loss of HRD1 results in elevated ENTPD5, CPT2, RMND1, and HSD17B4 protein levels and a consequent hyperactivation of both AMPK and AKT pathways. Genome-wide mRNA sequencing revealed that HRD1-deficiency reprograms liver metabolic gene expression profiles, including suppressing genes involved in glycogenesis and lipogenesis and upregulating genes involved in glycolysis and fatty acid oxidation. We propose HRD1 as a liver metabolic regulator and a potential drug target for obesity, fatty liver disease, and insulin resistance associated with the metabolic syndrome.

[1] Department of Pathology, Northwestern University Feinberg School of Medicine, Chicago, IL 60611, USA. [2] State Key Laboratory of Proteomics, Beijing Proteome Research Center, National Center for Protein Sciences (Beijing), Beijing Institute of Lifeomics, 102206 Beijing, China. [3] Department of Endocrinology, Peking Union Medical College Hospital, Chinese Academy of Medical Science, 100730 Beijing, China. [4] Department of Medical Psychology, Public Health Institute of Harbin Medical University, 150081 Harbin, China. [5] Division of Endocrinology, Metabolism and Molecular Medicine, Department of Medicine, Northwestern University Feinberg School of Medicine, Chicago, IL 60611, USA. [6] Department of Medicine, Northwestern University Feinberg School of Mdicine, Chicago, IL 60611, USA. [7] Department of Pharmacology and Toxicology, University of Arizona, Tucson, AZ 85721, USA. [8] Center for Molecular Medicine and Genetics, Department of Biochemistry, Microbiology, and Immunology, Wayne State University School of Medicine, Detroit, MI 48201, USA. These authors contributed equally: Juncheng Wei, Yanzhi Yuan. Correspondence and requests for materials should be addressed to J.W. (email: wangjian@bmi.ac.cn) or to F.H. (email: hefc@bmi.ac.cn) or to D.F. (email: fangd@northwestern.edu)

Metabolic syndrome is a cluster of conditions including obesity, hypertension, hyperlipidemia, hyperglycemia, and insulin resistance that increase the risk of type 2 diabetes, nonalcoholic fatty liver disease (NAFLD), and cardiovascular disease[1,2]. The prevalence of metabolic syndrome is reaching epidemic proportions, with a worldwide estimate of 1.5 billion each year[3,4]. Metabolic syndrome is associated with a dysregulation of catalytic activities or expression levels of the metabolic enzymes that play a central role in synthesis and/or consumption of glucose, lipid, and proteins. Metabolic enzyme activity is regulated by multiple mechanisms, including changes in substrate concentration, modulation of mRNA transcription, and activation or inhibition of rate-limiting enzymes at the posttranslational level[5,6]. The roles of highly responsive, energy saving, and reversible covalent posttranslational modifications of the metabolic enzymes, such as phosphorylation, acylation, and methylation, have been exhaustively explored[7–9]. In contrast, the contribution of proteolysis leading to key metabolic enzyme turnover, as well as the specific role of ubiquitin E3 ligases, to the regulation metabolic enzyme activity remains to be defined[10].

HMG-CoA reductase degradation protein 1 (HRD1) was identified as an E3 ligase to control cholesterol production through regulation of rate-limiting enzyme HMGCR (HMG-CoA reductase) turnover in yeast[11–13]. Consequent studies have shown that HRD1 plays a critical role in endoplasmic reticulum (ER)-associated degradation of misfolded/unfolded proteins and protects cells from ER stress-induced cell death[14]. In addition to misfolded protein substrates, HRD1 was found to catalyze ubiquitin-conjugation of IRE1α, an ER stress sensor, to inhibit cell apoptosis and control intestinal stem cell homeostasis[14]. The IRE1α downstream transcription factor Xbp-1 has been shown to regulate HRD1 mRNA transcription, providing an interesting feedback loop for HRD1-IRE1α during ER stress response[15]. HRD1 was re-named Synoviolin because it is upregulated in synovial fibroblasts in patients with rheumatoid arthritis[16], and its upregulation is induced by proinflammatory cytokines through a pathway distinct from IRE1α-Xbp-1[17]. We recently found that HRD1 is involved in immune regulation in dendritic cell antigen presentation, as well as in the activation of both T and B lymphocytes[18–20]. However, the organ-specific physiological and pathological roles of HRD1 remain largely uncharacterized.

In this study, we found that HRD1 was postprandially induced to control the metabolic balance in the physiological condition. Genetic HRD1 suppression specifically in the liver protected mice from high-fat diet (HFD)-induced obesity, hyperlipidemia, NAFLD, and insulin resistance. We further demonstrated that HRD1 serves as a positive effector of metabolism through direct binding and increase of the ubiquitination of key metabolic enzymes.

## Results

**HRD1$^{Alb}$ mice display the reduction of body weight and triglyceride (TG) levels**. It has been shown that HRD1 variants are associated with NAFLD[21]. Real-time quantitative PCR (qPCR) analysis detected higher expression levels of HRD1 in the mouse liver than in other organs, implying that HRD1 may play an important role in the liver (Fig. 1a). Further analysis of HRD1 expression in mice during fasting-refeeding conditions revealed that liver HRD1 expression, while not altered by fasting, was increased three- to five-fold during the early phase of refeeding, then returned to basal levels within 6 h after refeeding (Fig. 1b). In contrast, the expression of *Hrd1* was not affected by HFD because *Hrd1* mRNA levels are comparable between mice fed with normal chew and HFD (Supplementary Fig. 1a), implying

that liver HRD1 expression is regulated by food intake and that HRD1 likely plays an important role in metabolic regulation. To define HRD1 function in postprandial metabolism, we crossed HRD1 floxed (*HRD1$^{f/f}$*) mice with mice harboring a Cre transgene under the control of the albumin enhancer/promoter (Alb-Cre$^+$) and generated liver-specific HRD1 knockout (*HRD1$^{f/f}$*Alb-Cre$^+$; HRD1$^{Alb}$) mice (Supplementary Fig. 1b, c). Both HRD1 mRNA and protein expression were eliminated in the liver of HRD1$^{Alb}$ mice (Supplementary Fig. 1d, e). Interestingly, while body weights of *HRD1$^{f/f}$*Alb-Cre$^-$ (Control) and HRD1$^{Alb}$ mice were comparable at a young age, the HRD1$^{Alb}$ mice gained less body weight with age, with an average of 15% lower body weight compared to the control mice by the age of 12 weeks (Fig. 1c). Further analysis of blood glucose levels indicated that, while glucose levels under fasting conditions were indistinguishable between controls and HRD1$^{Alb}$ mice, a significant reduction in serum glucose levels was detected in HRD1$^{Alb}$ mice after refeeding, implying that hepatic HRD1-deficiency facilitates glucose consumption (Fig. 1d). Similarly, we detected dramatically lower serum TG levels in HRD1$^{Alb}$ mice compared to control mice (Fig. 1e).

To determine whether the reduction in body weight gain as well as serum glucose and TG in HRD1$^{Alb}$ mice was the result of reduced food intake and/or altered energy metabolism, we examined the metabolic phenotype of 12-week-old mice, the age at which the HRD1$^{Alb}$ mice showed a significant reduction in body weight gain. Interestingly, while their activity levels were comparable between control and HRD$^{Alb}$ mice, the average intake of both food and water by HRD1$^{Alb}$ mice was even significantly higher than that of the control mice (Fig. 1f, g). Consistent with this, energy expenditure, reflected by 24-h O$_2$ consumption and CO$_2$ production, were higher in HRD1$^{Alb}$ mice compared to that of the control mice (Fig. 1h, i). In addition, the respiratory exchange ratio (RER) was increased during the light period in the HRD1$^{Alb}$ mice (Fig. 1j). These data suggest that increased energy expenditure could be a cause of the reduced body weight gain in HRD1$^{Alb}$ mice.

As Alb-Cre mediates the deletion of *HRD1* gene from the early stages of liver development, the possibility that the reduced body weight gain and increased energy expenditure in HRD1$^{Alb}$ mice due to a liver developmental defect cannot be fully excluded. However, this is unlikely because liver sizes were comparable between the control and HRD1$^{Alb}$ mice. Hematoxylin and eosin (H&E) staining did not detect any histological abnormalities in the liver of HRD1$^{Alb}$ mice (Supplementary Fig. 2a). Further Oil Red O staining did not detect an increase in lipid deposition in the liver of both wild-type (WT) and HRD1$^{Alb}$ mice fed a standard chow diet (Supplementary Fig. 2b). More importantly, the serum aspartate transaminase, alanine transaminase, and insulin in the HRD1$^{Alb}$ mice were not increased (Supplementary Fig. 2c, d). Next, we generated a mouse strain with an inducible HRD1 deletion by breeding *HRD1$^{f/f}$* mice with mice harboring a Cre transgene under the control of an interferon-inducible myxovirus resistance 1 (Mx1) promoter. As indicated in Supplementary Fig. 1f, HRD1 mRNA was diminished in the liver of *HRD1$^{f/f}$*Mx1-Cre$^+$ mice after intraperitoneal injection of polyI:C. Consistent with our observation in HRD1$^{Alb}$ mice, a statistically significant reduction in the body weight gain in the *HRD1$^{f/f}$*Mx1-Cre$^+$ mice was detected 1 month after polyI:C injection (Fig. 1k). In addition, the basal levels of both cholesterol and TG, as well as glucose level, were dramatically decreased in the *HRD1$^{F/F}$*Mx1-Cre$^+$ mice after refeeding (Fig. 1l, m). These results exclude the possibility that the elevated energy expenditure in HRD1$^{Alb}$ mice is due to a liver developmental defect. To further support this conclusion, the adenovirus-Cre-mediated *HRD1* deletion in *HRD1$^{f/f}$* mice also reduced the glucose level

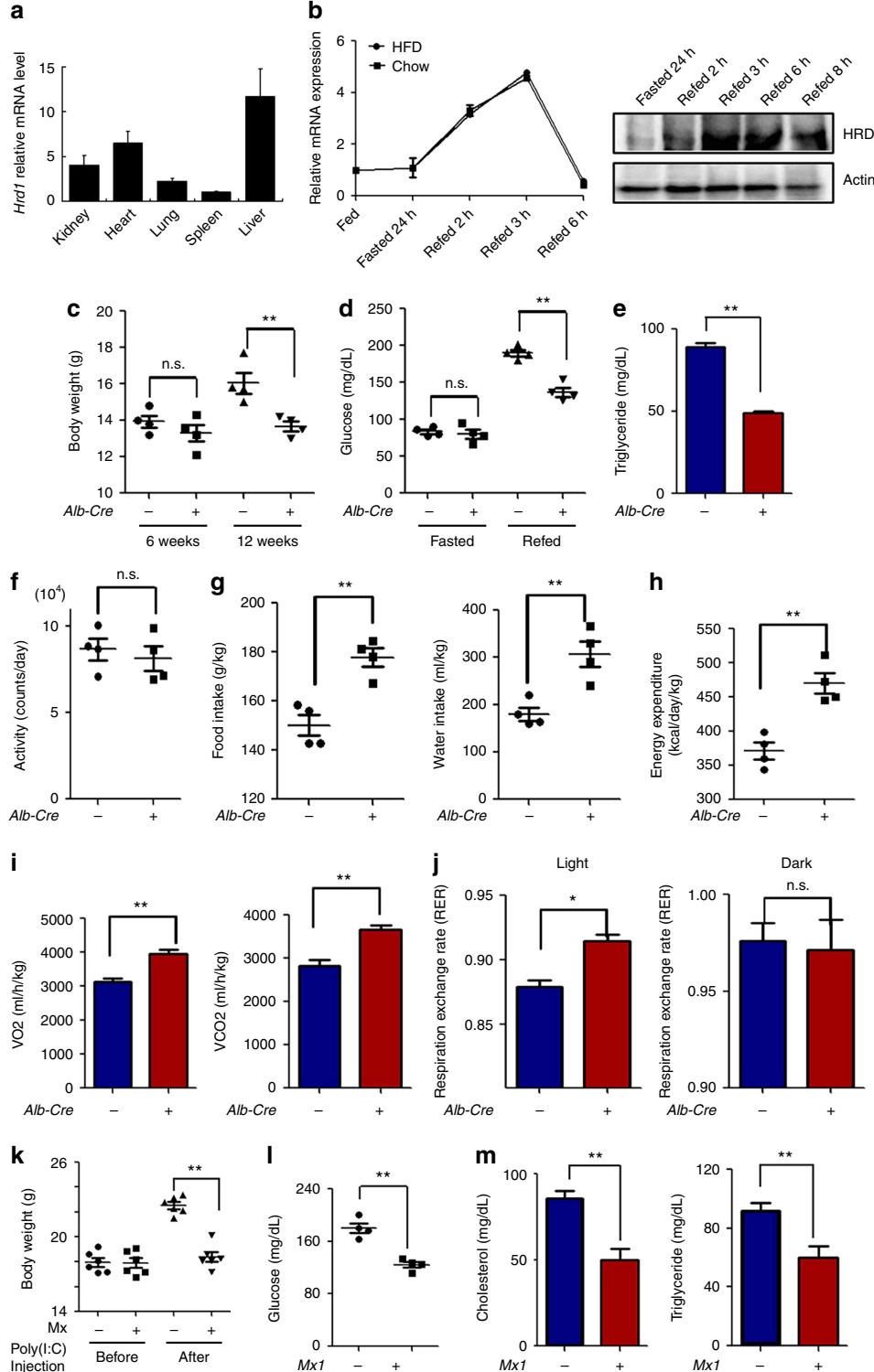

**Fig. 1** Mice with liver-specific HRD1 deletion display the reduction of body weight, lipid, and glucose. **a** *Hrd1* mRNA levels in different organs. **b** *Hrd1* mRNA and protein levels under fasted and refed conditions. **c** Body weights of WT and HRD1^Alb mice at weeks 6 and 12 ($n = 4$ for each group). **d** Blood glucose levels of WT and HRD1^Alb mice under fasted and refed conditions ($n = 4$ for each group). **e** Serum TG of WT and HRD1^Alb mice under refed conditions ($n = 6$ for each group). **f** Activities of WT and HRD1^Alb male mice ($n = 4$ for each group). **g** Food intake, water intake, and body weight ratios of WT and HRD1^Alb mice ($n = 4$ for each group). **h** Energy expenditure and body weight ratios of WT and HRD1^Alb mice ($n = 4$ for each group). **i** $O_2$ consumption and $CO_2$ production of WT and HRD1^Alb mice fed a standard chow diet ($n = 8$ for each group). **j** RER of WT and HRD1^Alb mice fed a standard chow diet ($n = 8$ for each group). **k** Body weight of the *HRD1^{F/F} and HRD1^{F/F}*Mx1-Cre^+ mice ($n = 6$ for each group) before and 1 month after polyI:C injection. **l** Blood glucose was measured 7 days after polyI:C injection in the refed condition. **m** Serum cholesterol and TG level were measured 7 days after polyI:C injection under refed conditions ($n = 6$ for each group). Data are representative of three independent experiments (mean ± s.d.). *$P < 0.05$. **$P < 0.01$ by unpaired Student's *t* test

even after refeeding (Supplementary Fig. 3b, c). Therefore, HRD1 is a liver autonomous regulator that controls liver metabolism.

**HRD1 deletion protects mice from HFD-induced obesity**. To study the role of HRD1 in hepatic metabolism and obesity, HRD1[Alb] mice and their littermate controls were placed on a HFD at the age of 6 weeks, whereas the body weights of HRD1[Alb] and control mice were indistinguishable (Fig. 1c). As expected, the control mice gradually gained body weight on the HFD, but the body weight gain of HRD1[Alb] mice was significantly reduced (Fig. 2a), indicating that HRD1 deletion protects mice from HFD-induced obesity. Further, the liver size and weight, as well as the accumulation of both subcutaneous and epidermal adipose tissue, were dramatically lower in HRD1[Alb] mice compared to WT mice (Fig. 2b, c). Histological analysis demonstrated that the liver of WT but not of HRD1[Alb] mice developed severe steatosis after 14 weeks on HFD (Fig. 2d). Staining with Oil Red O confirmed the presence of massive lipid deposition in the liver of control mice, but not in that of HRD1[Alb] mice (Fig. 2e). These results clearly indicate that liver-specific HRD1 deletion protects mice from HFD-induced obesity and hepatic steatosis.

It is well established that a long-term HFD induces NAFLD and insulin resistance. Indeed, WT mice on the HFD showed impaired glucose tolerance and insulin sensitivity compared to HFD-fed HRD1[Alb] mice (Fig. 2f, g). In addition, serum TG and cholesterol levels were much lower for HRD1[Alb] mice while on a HFD compared to control littermates (Fig. 2h). HFD often leads to the elevated expression of SREBP1 and SCD1, both of which are lipogenic genes corresponding to increased liver fat storage[22,23]. Interestingly, both *Scd1* and *Srepf1* expression levels were significantly reduced in the liver of HFD-fed HRD1[Alb] mice compared to controls (Fig. 2i). The reduction in the hepatic lipid accumulation and insulin resistance in the HRD1[Alb] mice were not due to decreased food intake on the HFD, as the average food intake by HRD1[Alb] mice was actually significantly higher than the WT controls, while activity levels were not different (Fig. 2j). This excluded the possibility that liver HRD1 deletion protects mice from obesity by decreasing food intake and increasing mouse activity. Consistent with our observation in mice fed a standard chow diet, HFD-fed HRD1[Alb] mice displayed increased energy expenditure (Fig. 2k). To support this notion, we observed the increased 24-h $O_2$ consumption and $CO_2$ production by HFD-fed HRD1[Alb] mice (Supplementary Fig. 4a, b). As a consequence, the RER profile of the HFD-fed HRD1[Alb] mice was also increased in the dark hours (Supplementary Fig. 4c, d). These results indicate that liver-specific HRD1 deletion protects against HFD-induced obesity and fatty liver disease through improvement in metabolic activity.

**HRD1 interacts with key metabolic regulatory proteins**. As an E3 ligase, HRD1 may regulate liver metabolism through ubiquitination and degradation of proteins involved in metabolic activity[24]. First, we purified and analyzed HRD1-binding proteins from human liver HepG2 cells using affinity purification coupled to mass spectrometry (AP-MS) (Fig. 3a). We specifically pulled down endogenous HRD1 protein using an anti-HRD1 antibody from HepG2 cells, as confirmed by immunoblotting (Supplementary Fig. 5a). MS characterization of the proteins in complex with endogenous HRD1 yielded a total of 347 proteins from three independent experiments (Supplementary Fig. 5b, iProx access id: IPX0000928000). The reliability analysis using two different computational scoring procedures, COMPASS and SAINT, revealed a total of 75 proteins (40% overlap rate) with a *P* value of <0.01 (Fig. 3b). Further analysis of the HRD1 and control samples by correlation matrix revealed a very strong correlation between

the three biological replicates (average correlation coefficient of 0.86), while their correlation with controls was rather modest (average correlation coefficient is 0.3) (Fig. 3c). Therefore, our AP-MS approach identified with high confidence HRD1-specific binding proteins. Analysis of the biological processes identified a subgroup of proteins important for the ER unfolded protein response, some of which had previously been shown to be associated with HRD1[25,26], further validating our AP-MS results. Importantly, a large portion of the HRD1-interacting proteins are involved in a variety of metabolic regulatory processes, including fatty acid (FA) transportation, ATP metabolic processes, and lipid and glucose metabolism (Fig. 3d and Supplementary Fig. 6). Further analysis indicated that HRD1-binding proteins localize at membrane-bounded organelles, such as the ER. Interestingly, 17 HRD1-interacting proteins were identified to localize at the mitochondria, suggesting that HRD1 may serve as an E3 ligase in mitochondrial protein quality control (Fig. 3e). These results suggest that HRD1 controls liver metabolism through regulation of multiple metabolic signaling pathways.

Importantly, the interaction of HRD1 with five novel HRD1 interactors—ENTPD5, CPT2, RMND1, HSD17B4, and ATP5D— were confirmed in transiently transfected HEK293 cells by co-immunoprecipitation (Co-IP) and immunoblotting (Fig. 3f). The endogenous ENTPD5, CPT2, RMND1, and HSD17B4 proteins were also detected in anti-HRD1 immunoprecipitates but not in normal rabbit immunoglobulin G (IgG) controls, confirming their endogenous interactions (Fig. 3g). ENTPD5 is part of an ATP hydrolysis cycle that converts ATP to AMP, resulting in a compensatory increase in glycolysis[27]. Both HSD17B4 and CPT2 are multifunctional enzymes important for FA oxidation[28,29]. In addition, RMND1 is likely involved in mitochondrial metabolic activation[30]. Therefore, HRD1 may control liver metabolism through targeting specific metabolic regulators, including ENTPD5, CPT2, RMND1, HSD17B4, and ATP5D. The HRD1 protein contains a six transmembrane domain at the N-terminus that recognizes misfolded proteins, an E3 ligase catalytic RING finger domain, and a proline-rich domain at the C-terminus that directly interacts with proteins localized in the cytoplasm[20,24]. Using the truncated HRD1 mutants, we further demonstrated that the C-terminus of HRD1 directly mediates interaction with ENTPD5, CPT2, and RMND1 (Fig. 3h).

**HRD1 is an E3 ubiquitin ligase for liver metabolic regulators**. An E3 ubiquitin ligase catalyzes ubiquitin conjugation to proteins targeted for proteasomal and/or liposomal degradation[17]. Indeed, we detected an average two- to five-fold increase in the levels of ENTPD5, HSD17B4, CPT2, and RMND1 protein in the liver of HRD1[Alb] mice compared to WT controls (Fig. 4a). In contrast, qPCR analysis confirmed that mRNA levels of ENTPD5, HSD17B4, CPT2, and RMND1 are unaltered by HRD1 gene deletion in the liver (Supplementary Fig. 7), implying that HRD1 regulates the expression of these four critical metabolic regulators at a posttranscriptional level. We previously reported that IRE1α stability is controlled by HRD1 in synovial fibroblasts and intestinal endothelial cells[14,17]; here we similarly found that IRE1α protein but not mRNA was elevated in HRD1[Alb] liver tissues (Fig. 4a). CPT2 and RMND1 are necessary for lipid and protein transportation[31]. HSD17B4 is important for the FA oxidation[32]. ENTPD5, together with CMPK1 and AK1, constitute an ATP hydrolysis cycle that converts ATP to AMP, resulting in a compensatory increase in glycolysis[27]. Previous studies have shown that IRE1α is a key factor involved in silencing lipid metabolism genes and lowering plasma lipid levels in mice[33]. Therefore, our results suggest that HRD1 regulates liver metabolism through degradation of critical regulators involved in

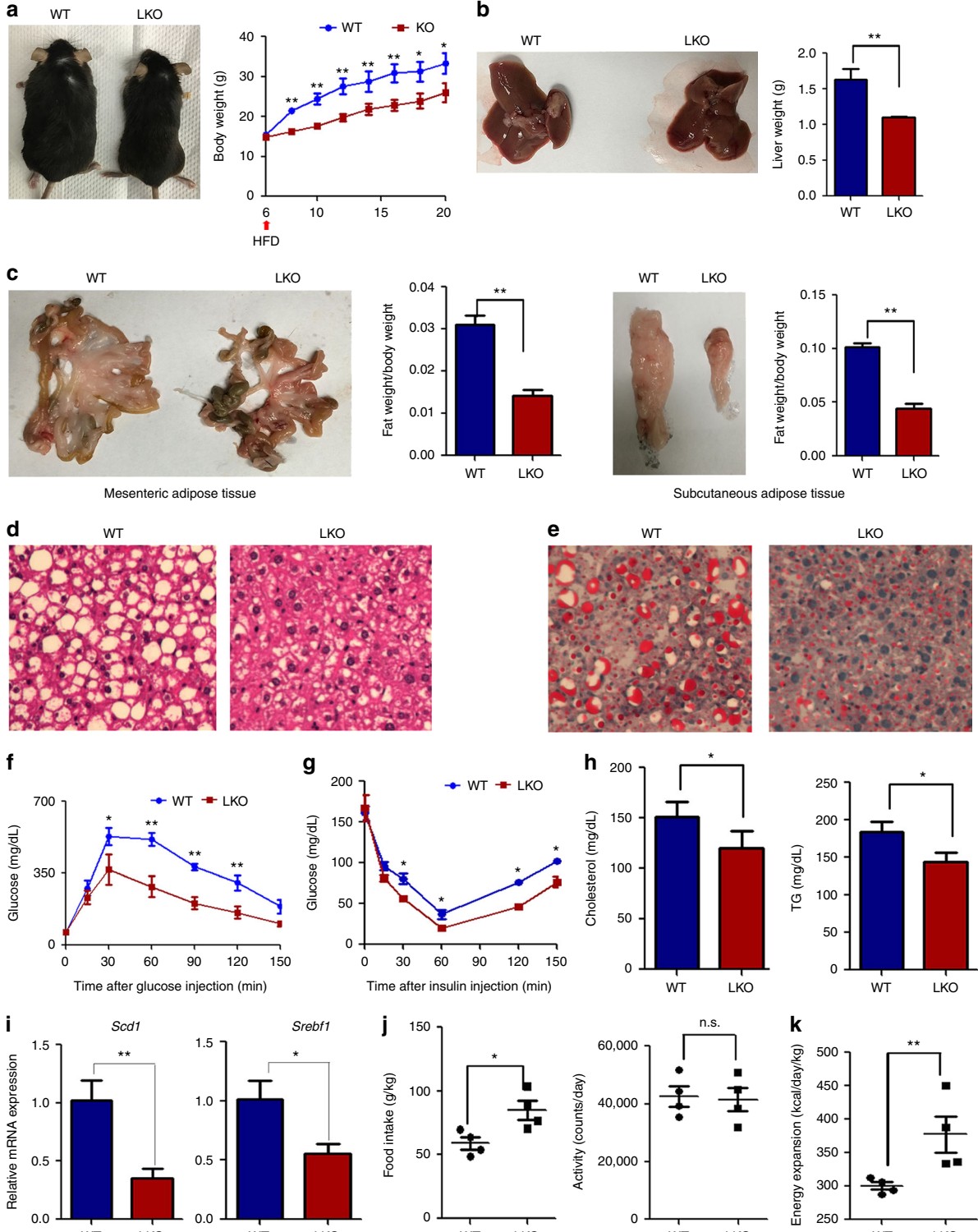

**Fig. 2** HRD1 inhibition protects mice from HFD-induced obesity and fatty liver disease. **a** Body weights of WT and HRD1[Alb] mice fed a HFD were measured every 2 weeks. Liver (**b**) and fat (**c**) weights from WT and HRD1[Alb] mice fed a HFD for 14 weeks (n = 6 for each group). H&E (**d**) and Oil Red O (**e**) staining of the liver from **b** (n = 6 for each group). Glucose tolerance assay (**f**) and insulin tolerance assay (**g**) were performed on mice after 14 weeks of a HFD (n = 6 for each group). **h** Serum TG and cholesterol levels from WT and HRD1[Alb] mice after 14 weeks of a HFD under the refed condition (n = 6 for each group). **i** Hepatic gene expression of Srebp1 and Scd1 in WT and HRD1[Alb] mice after 14 weeks of a HFD under the refed condition (n = 6 for each group). **j** Food intake, body weight ratios, and activities of WT and HRD1[Alb] mice after 14 weeks of a HFD (n = 4 for each group). **k** Energy expenditure of WT and HRD1[Alb] mice after 14 weeks of a HFD (n = 4 for each group). The data are representative of three independent experiments (mean ± s.d.). *P < 0.05. **P < 0.01 by unpaired Student's t test

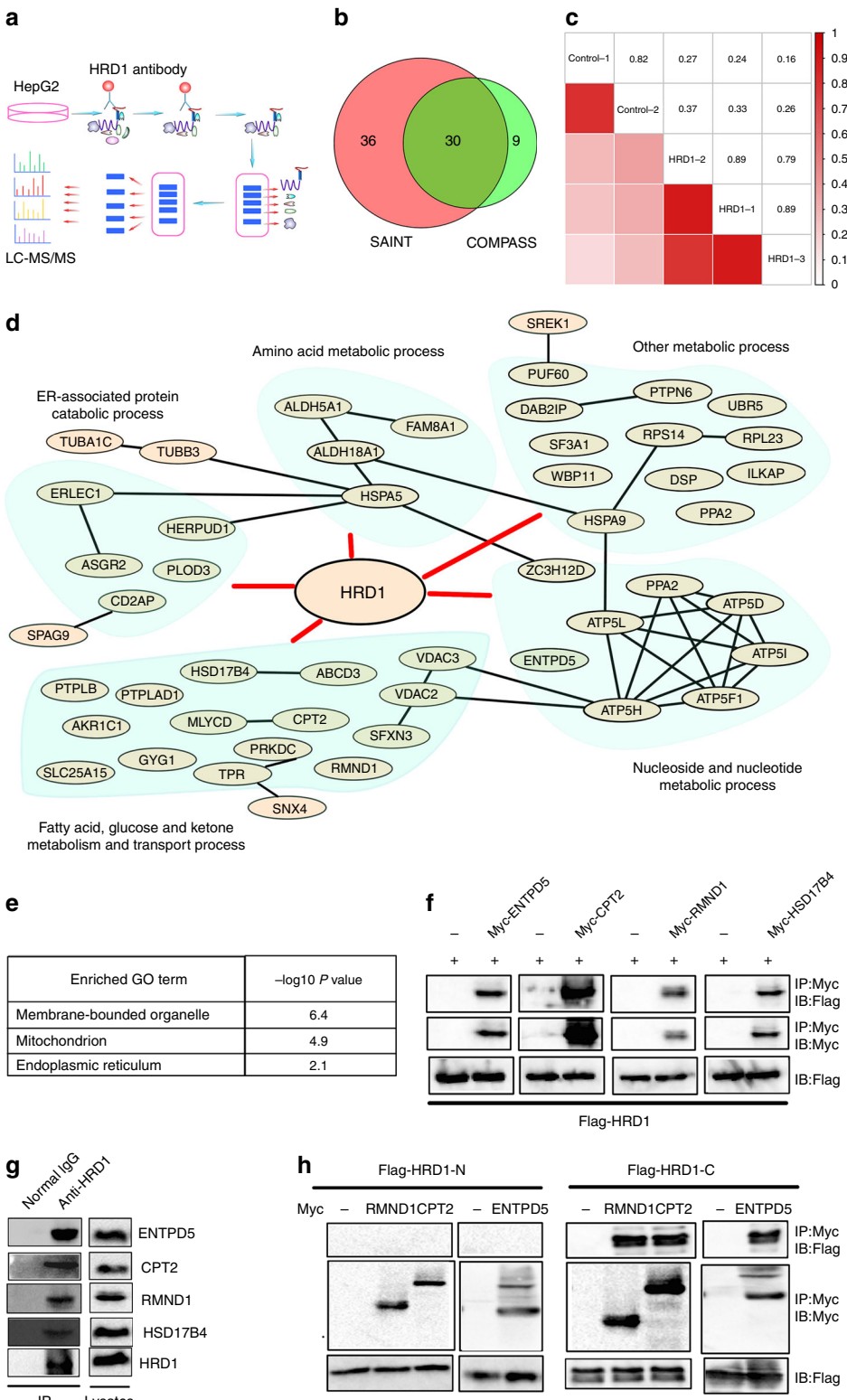

**Fig. 3** Identification of potential E3 ligase HRD1-interacting protein substrates by AP-MS. **a** Flow chart for the proteomic identification of HRD1-binding proteins. **b** Venn diagram of high-confidence proteins derived from the COMPASS and SAINT software. **c** A correlation matrix based on spectral counts of each protein was constructed for the three repeats of HRD1 purification and two repeats of control samples using normal IgG. Pairwise Pearson correlation scores were measured. **d** Biological process analysis of the functions of HRD1-binding proteins. **e** GO localization analysis of the HRD1-binding proteins. **f** Validation of the interactions between HRD1 and potential HRD1-binding proteins from proteomic screening. **g** Endogenous interaction between HRD1 and screened proteins from **f**. **h** C-terminus of HRD1 mediates the interaction with CPT2, RMND1, and ENTPD5. Data are representative of three independent experiments

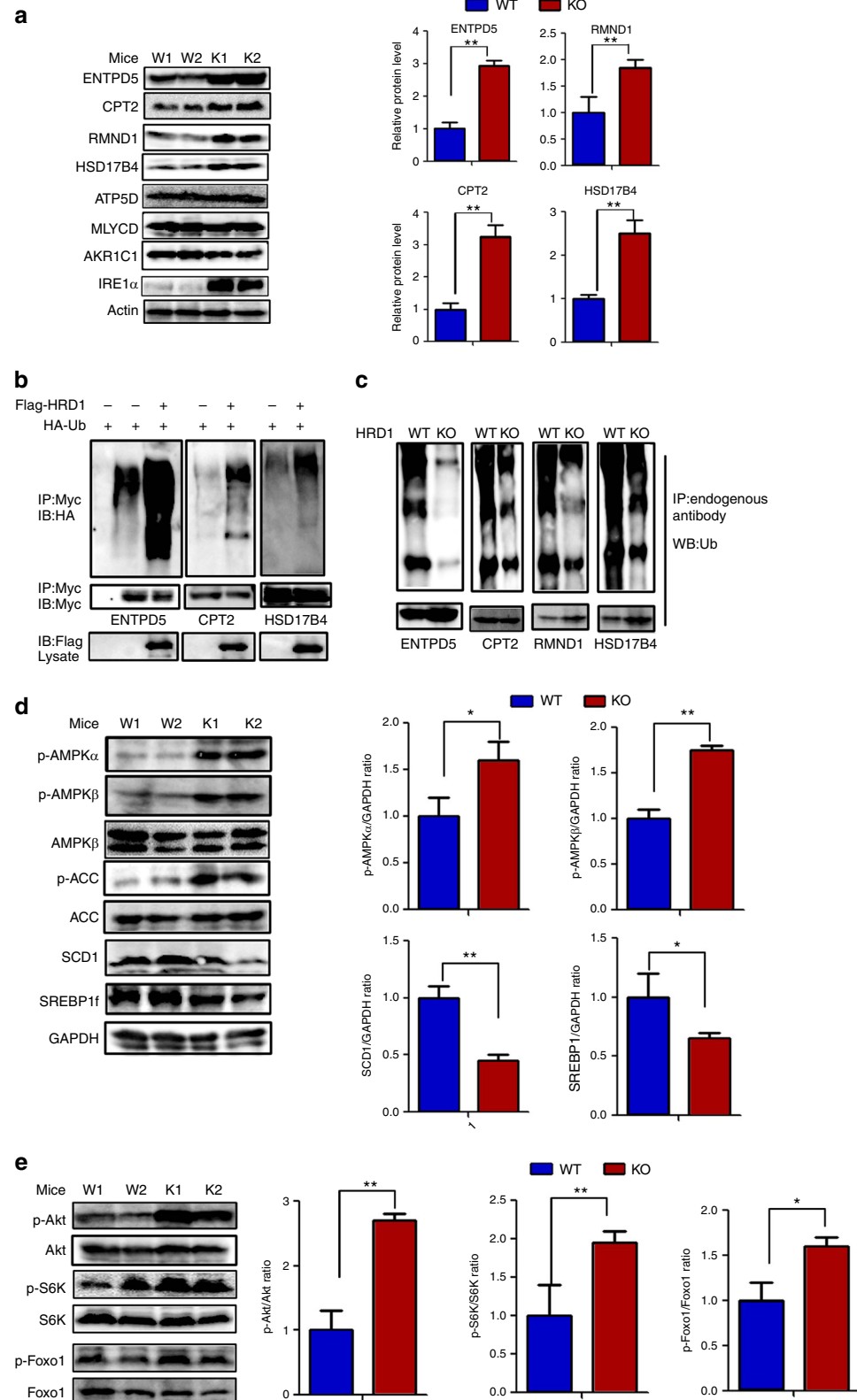

**Fig. 4** HRD1 is an E3 ubiquitin ligase for liver metabolic regulators. **a** HRD1-binding protein levels in the liver of WT and HRD1^Alb mice under the refed condition. **b** Ubiquitination of HRD1 substrates after co-transfection with HRD1 in 293T cells. **c** Ubiquitination of HRD1 substrates in the liver of WT and HRD1^Alb mice under the refed condition. **d** Immunoblots of hepatic AMPK and AMPK targets in the WT and HRD1^Alb mice under the refed condition. **e** Immunoblots of hepatic p-Akt, Akt, p-S6K, S6K, p-Foxo1, and Foxo1 levels in the WT and HRD1^Alb mice under the refed condition. Data are representative of three independent experiments (mean ± s.d.). *$P < 0.05$. **$P < 0.01$ by unpaired Student's $t$ test

several metabolic signaling pathways. Of note, the levels of some other proteins identified as interactors of HRD1, such as ATP5D, MLYCD and AKR1C1, were not affected by HRD1 deletion in the liver (Fig. 4a).

We next asked whether HRD1 promotes ubiquitination of ENTPD5, HSD17B4, and CPT2, all of which protein expression levels were dramatically increased in HRD1-null liver. Indeed, ectopic HRD1 expression significantly enhanced the ubiquitin conjugation of ENTPD5, HSD17B4, and CPT2 (Fig. 4b). Conversely, ubiquitination levels of ENTPD5, CPT2, and RMND1 and HSD17B4 were dramatically decreased in the liver of HRD1[Alb] mice compared to control mice (Fig. 4c). These results indicate that HRD1 is an E3 ubiquitin ligase of ENTPD5, CPT2, and RMND1 and HSD17B4, all of which play critical roles in liver metabolic regulation.

**AMP-activated protein kinase (AMPK) signaling pathways are upregulated in the liver of HRD1[Alb] mice**. It has been shown that ENTPD5 converts ATP to AMP to increase the AMP/ATP ratio and activate AMPK gluconeogenesis and lipogenesis suppression[27,34,35]. Our discovery that HRD1 catalyzes ENTPD5 ubiquitination for degradation suggests that HRD1 may regulate the AMPK pathway to control liver metabolism. Indeed, a significantly higher level of phosphorylated AMPKα and AMPKβ was detected in the liver of HRD1[Alb] mice compared to WT controls. As a consequence, phosphorylation of ACC1, a well-defined AMPK substrate, was also higher in HRD1[Alb] mice (Fig. 4d). As a consequence, significantly lower expression levels of SCD1 and SREBF1 were detected in HRD1[Alb] mice (Fig. 4d). As ENTPD5 is also shown to activate the phosphoinositide-3 kinase/Akt pathway[36], we the asked whether HRD1-mediated ENTPD5 protein destruction affects glucose metabolism through the insulin-Akt signaling pathway. Consistent with this hypothesis, we observed that the phosphorylation, but not total protein expression levels, of Akt, as well as its downstream targets Foxo1 and S6K, were significantly elevated in HRD1[Alb] liver compared to control liver tissue, indicating that HRD1 deletion enhances the activation of the insulin signaling pathway in liver (Fig. 4e).

**HRD1 deletion reprograms liver metabolic gene expression profiles**. To further investigate the role of HRD1 in liver metabolic regulation, we analyzed the mRNA expression profile in the liver from fasted and refed HRD1[Alb] and WT mice. Under the fasted condition, the expression of 158 genes was decreased by >50% and the expression of 601 genes was increased at least two-fold by HRD1 deletion (Fig. 5a). After refeeding, the RNA expression profile was dramatically altered, with the expression of 567 genes decreased by >50% and 1204 genes increased by more than two-fold in the HRD1[Alb] liver (Fig. 5a, b). Functional analysis indicated that most of the genes altered by HRD1 deletion are involved in metabolic regulation, cell growth, differentiation, and survival, indicating that HRD1 plays a fundamental role in a variety of critical biological processes (Fig. 5c). Consistent with our previous observations in immune cells, deletion of HRD1 caused increases in the expression of only few ER stress-response genes such as Ddit3 (Chop), Trib3, and Gadd45 but not Xbp1s (Fig. 5a), which are downstream targets of the PKR-like ER kinase (PERK)-ATF4 branch of the ER stress pathways important for the lipogenic gene expression[37]. However, administration of a PERK-specific inhibitor to the HRD1[Alb] mice, despite the Ddit3 mRNA expression was dramatically inhibited, did not alter the mRNA expression of Srebf1, Chrebf1, Scd1, Fasd2, Gyk, Agpat2, Apob, and Tymp (Supplementary Fig. 9), largely excluding the possibility that HRD1 regulates the lipid metabolism through activation of the PERK-ATF4 pathway.

Functional analysis of the genome-wide RNA sequencing (RNA-seq) data identified that a large proportion of the altered genes important for metabolic processes, including glucose, FA and lipid, amino acid, and nucleotide metabolism (Fig. 5c, Gene Expression Omnibus (GEO) access number: GSE113154). Further functional enrichment analysis showed that HRD1 deletion resulted in significant changes in the expression of a large number of genes critical for lipid and FA metabolism (Fig. 5c). These results indicate that HRD1 is a critical regulator in liver metabolism. Both AMPK and AKT pathways suppress lipogenesis through suppressing downstream target genes, including Srebf1 and Chrebp1[7]. Consistent with our discovery that HRD1 deletion led to elevated ENTPD5 expression, as well as AMPK and AKT activation, the expression of transcription factors Srebf1 and Chrebp1, as well their target genes Scd1, Fads2/6, and Elovl2/3, were dramatically decreased in our RNA-seq analysis (Fig. 5d), which was validated by qPCR (Fig. 5e, f). Genes that control FA and TG synthesis (Fig. 5f, g), Scd1, Fasn, Fads2, Gyk, and Agpat2/6, were decreased and genes involved in FA oxidation, Acot2, Acot9, Fabp7, and Lpl, were increased in the livers by HRD1 deletion (Supplementary Fig. 8). Genes that regulate amino acid synthesis, Upp2 and Tymp, were also decreased by HRD1 deletion (Fig. 5h). Therefore, targeted HRD1 deletion resulted in a shift in the metabolic gene expression profile in the mouse liver.

Interestingly, our RNA-Seq data showed that HRD1 deletion increased the Fgf21 expression. We confirmed the elevation of Fgf21 mRNA by reverse transcriptase–qPCR methods, and we also found that the fibroblast growth factor 21 (FGF21) protein levels in the serum were increased after hepatic HRD1 deletion (Supplementary Fig. 10a, b). The elevation of energy expenditure of the HRD1[Alb] mice was rescued by FGF21 deletion (Supplementary Fig. 10c). However, deletion of FGF21 failed to rescue the decreased levels of serum TG and FA in HRD1[Alb] mice and stearoyl-CoA desaturase (Scd1) expression (Supplementary Fig. 10d, e), an enzyme that catalyzes the rate-limiting step in the formation of monounsaturated FA, suggesting that HRD1 may regulate TG and FA metabolism through alternative pathways in the absence of FGF21. To further support this conclusion, Hrd1 deletion sufficiently protected mice from the HFD-induced fatty liver disease even with Fgf21 deletion (Supplementary Fig. 11).

**HRD1 regulates liver metabolism partially through ENTPD5–AMPK axis**. To investigate whether the lower expression of genes involved in FA and TG synthesis in the HRD1-deficient liver are due to lower body weight of the HRD1[Alb] mice, we injected polyI:C into the Mx1-Cre HRD1[flox/flox] mice and their WT littermates to silence HRD1 expression in the liver. As shown in Fig. 6a, 2 days after polyI:C treatment when the mice body weight was unaltered, the expression levels of genes involved in FA synthesis including Srebf1, Chrebf1, and Scd1 were dramatically decreased in the Mx1-Cre HRD1[flox/flox] mice, indicating that these changes are not due to growth retardation. However, we are aware that, even without body weight reduction, the possibility that AMPK overactivation and lower expression of genes are due to the altered systemic metabolic factors such as serum FA levels cannot be fully excluded. We then detected p-AMPK and p-AKT in the in vitro cultured primary hepatocytes in which the culture media are synchronized. Consistent with our initial observation, AMPK but not AKT activation was significantly increased by HRD1 deletion (Fig. 6b). In addition, the expression of genes involved in FA synthesis including Srebf1, Chrebf1, Scd1, and Fads2 were dramatically decreased in HRD1 knockout (KO) primary hepatocytes (Fig. 6c). Collectively, our data indicate that liver HRD1 deletion affects AMPK activation and metabolic response in a

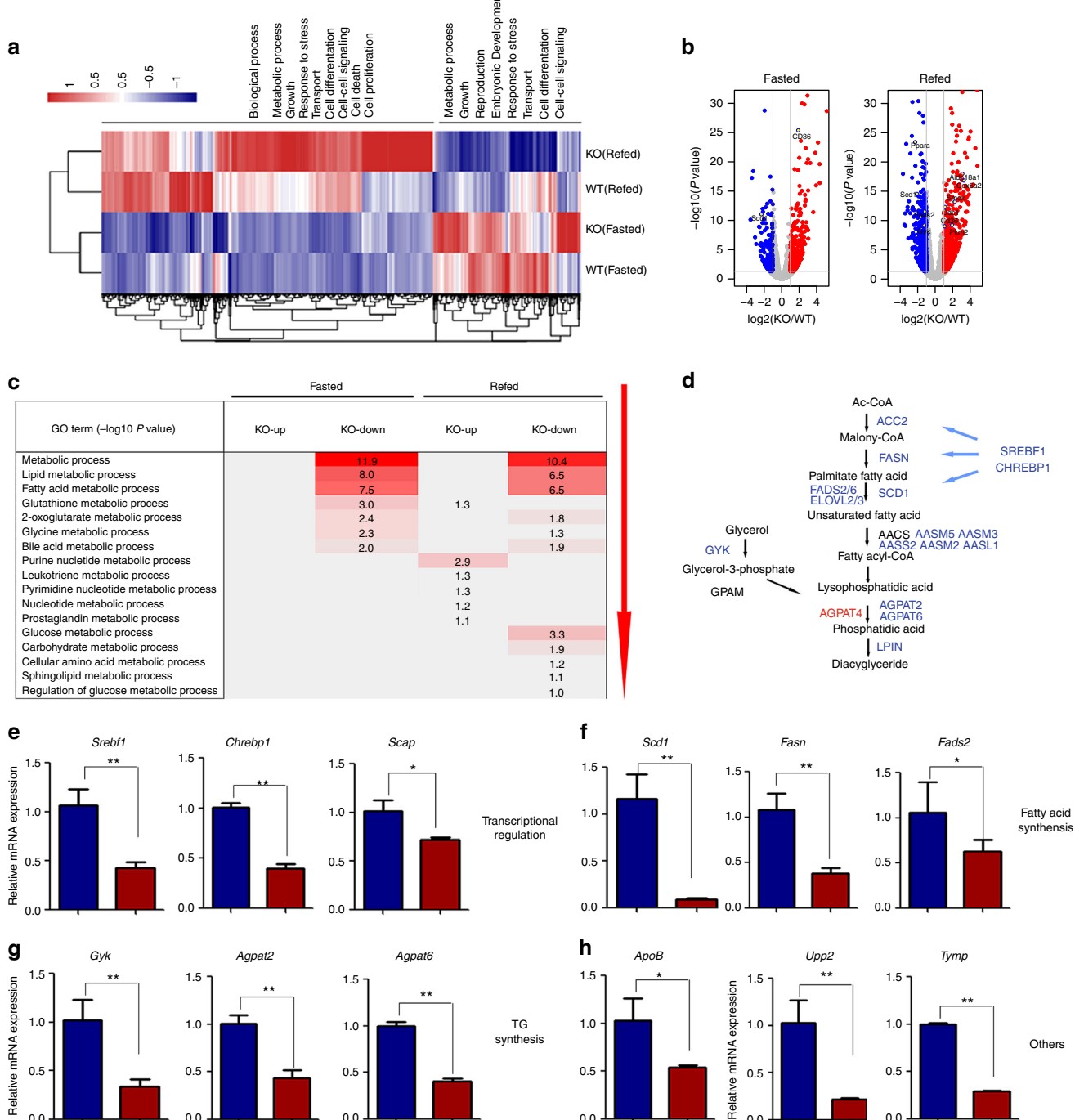

**Fig. 5** HRD1 deletion reprograms liver metabolic gene expression profiles. **a** Heatmap of upregulated or downregulated hepatic genes from WT and HRD1[Alb] mice. **b** Volcano plot of the genes from **a** under fasted and refed conditions. **c** Metabolic functions of the genes. **d** Key gene list related to metabolic function. **e–h** Hepatic gene expression profiles in WT and HRD1[Alb] mice under the refed condition ($n = 6$ for each group). The data are representative of three independent experiments (mean ± s.d.). *$P < 0.05$. **$P < 0.01$ by unpaired Student's $t$ test

hepatocyte-intrinsic manner. To further support this notion, AMPK inhibitor injection significantly increased the expression of genes involved in lipogenesis including *Chrebp1*, *Scd1*, and *Gyk* in HRD1[Alb] mice. Similarly, upregulation of the genes regulating the FA oxidation (*Lpl* and *Acot9*) in HRD1[Alb] mice was reversed by AMPK inhibitor administration (Fig. 6d and Supplementary Fig. 12). Therefore, HRD1 regulates liver metabolism through, at least partially, the ENTPD5-AMPK pathway.

Our data indicate that HRD1 functions as an E3 ubiquitin ligase of ENTPD5, an enzyme known to activate AMPK through

converting ATP to AMP[27]. To further determine whether the liver-specific HRD1 deletion alters liver metabolic response through the elevated ENTPD5, we infected primary hepatocytes from WT and HRD1 KO mice with lentivirus expressing *Entpd5*-specific short hairpin RNA (shRNA). *Entpd5*-specific shRNA efficiently knocked down ENTPD5 mRNA without altering *Hrd1* mRNA expression (Fig. 6e and Supplementary Fig. 13). Importantly, suppression of ENPTD5 expression partially rescued *Chrebp1* and *Scd1* expression in HRD1-null hepatocytes. In addition, the elevated *Lpl* expression in HRD1-null liver cells was

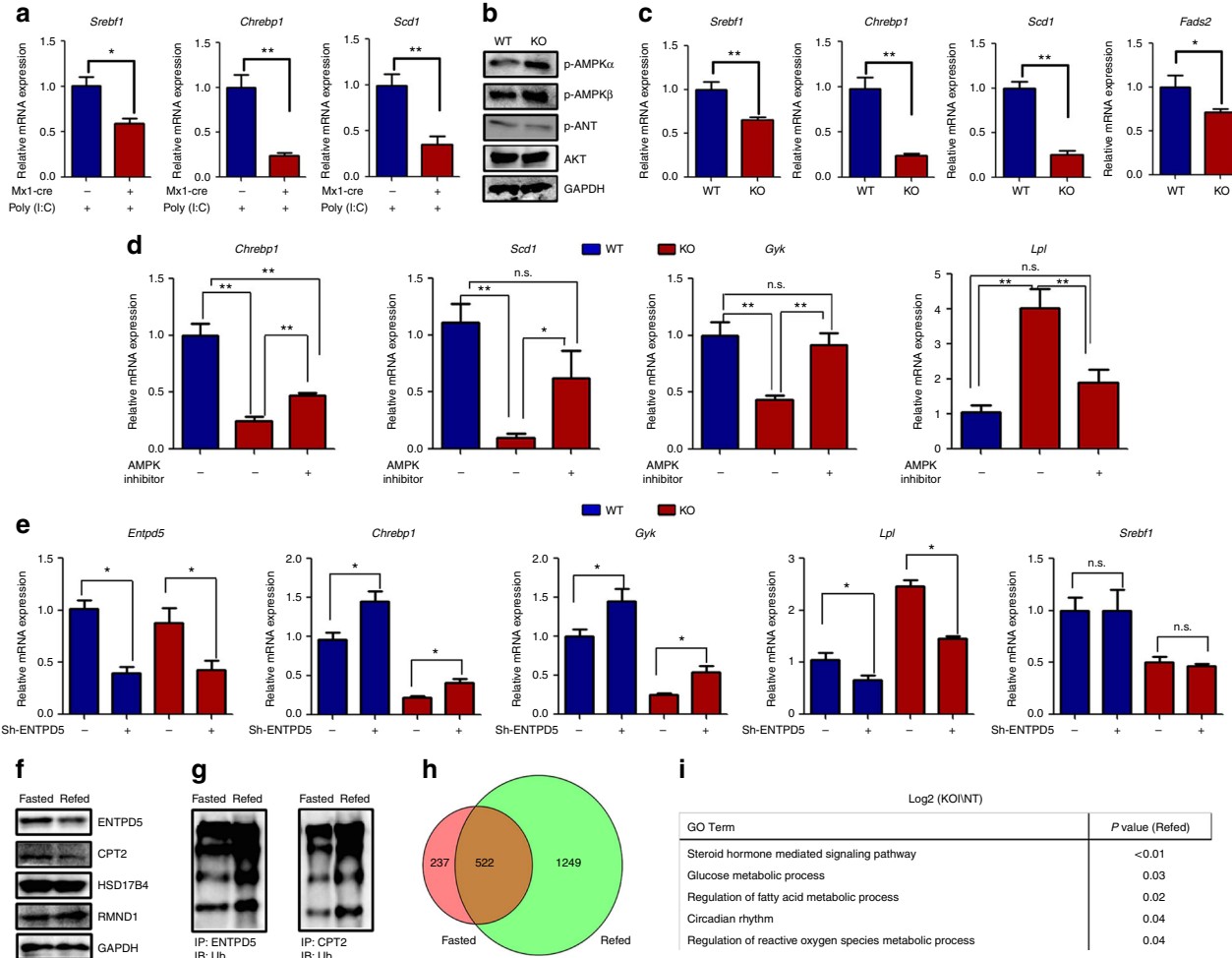

**Fig. 6** HRD1-mediated ubiquitination reprograms liver metabolic response partially through ENTPD5-AMPK pathway overactivation. **a** Relative mRNA of *Srebf1*, *Chrebp1*, and *Scd1* of HRD1[flox/flox] (WT) and HRD1[flox/flox] Mx1-Cre[+] mice 2 days after polyI:C injection ($n = 5$ for each group). **b**, **c** WT and HRD1[Alb] hepatocytes were isolated and cultured overnight. mRNA levels of *Srebf1*, *Chrebp1*, *Scd1*, and *Fads2* (**b**) and AMPK and AKT activity were measured (**c**). **d** WT and HRD1[Alb] mice were fasted overnight and refed for 1 h and then administered with AMPK inhibitor for additional 3 h. Hepatic mRNA levels of *Chrebp1*, *Scd1*, *Gyk*, *Srebf1*, *Lpl*, and *Acot9* were measured ($n = 5$ for each group). **e** WT and HRD1[Alb] hepatocytes were infected with lentivirus to specifically knockdown *Entpd5*. Two days after infection, hepatic mRNA levels of *Entpd5*, *Hrd1*, *Chrebp1*, *Gyk*, *Lpl*, and *Srebf1* were measured ($n = 5$ for each group). **f** Hepatic ENTPD5, CPT2, RMND1, and HSD17B4 protein levels were measured in the fasted and refed conditions. **g** Ubiquitination levels of ENTPD5 and CPT2 were measured in the fasted and refed conditions. **h** Differential genes in the fasted and refed genes were overlapped. **i** Metabolic functions of the genes from **h**. Data are representative of three independent experiments (mean ± s.d.). *$P < 0.05$. **$P < 0.01$ by unpaired Student's $t$ test

largely reversed by *Entpd5* knockdown. These results indicated that metabolic alteration induced by HRD1 deletion partially through the ENTPD5 accumulation.

**HRD1 reprograms liver metabolic response in mice upon refeeding.** Our observation that HRD1 gene transcription is induced by refeeding in mouse liver (Fig. 1b, c), implying a possible molecular mechanism by which refeeding induced HRD1 expression to regulate liver metabolism through catalyzing its targets for ubiquitination-mediated degradation. Indeed, a significant reduction in the protein expression of both ENTPD5 and CPT2 was detected in the liver tissue from refed mice (Fig. 6f). This reduction is likely due to the ubiquitination catalyzed by elevated HRD1 because the ubiquitination of ENTPD5 and CPT2 in the liver of refed mice was increased (Fig. 6f). However, while HRD1 deletion results in elevated HSD17B4 and RMND1 protein expression, a 4-h refeeding did not lead to a dramatic reduction of their protein expression levels, which is possibly due to a longer half-life or a different dynamic in ubiquitination-mediated

degradation HSD17B4 and RMND1 (Fig. 6g). Moreover, analysis of the gene expression profiles in the liver WT and HRD1[Alb] mice indicate that a substantial number genes was dramatically increased in HRD1-null liver only in the refed condition, and much less genes were altered in fasted condition (Fig. 6h and Supplementary Fig. 14, 1249 vs 237). Gene Ontology function analysis indicate that these genes are critical for multiple liver metabolic response pathways including steroid hormone-mediated signaling pathway, glucose and lipid metabolic process, and circadian rhythm (Fig. 6i). Collectively, our data suggest that HRD1-mediated ubiquitination plays an important role in reprogramming liver metabolic response in mice upon refeeding.

**HRD1 is a potential therapeutic target for HFD-induced obesity.** Our findings suggest that HRD1 may be a novel therapeutic target for liver metabolic disorders, including obesity and fatty liver disease. To test this hypothesis, we tested whether induction of HRD1 deletion in *HRD1[f/f]*Mx1-Cre mice liver 6 weeks after HFD feeding could suppress HFD-induced

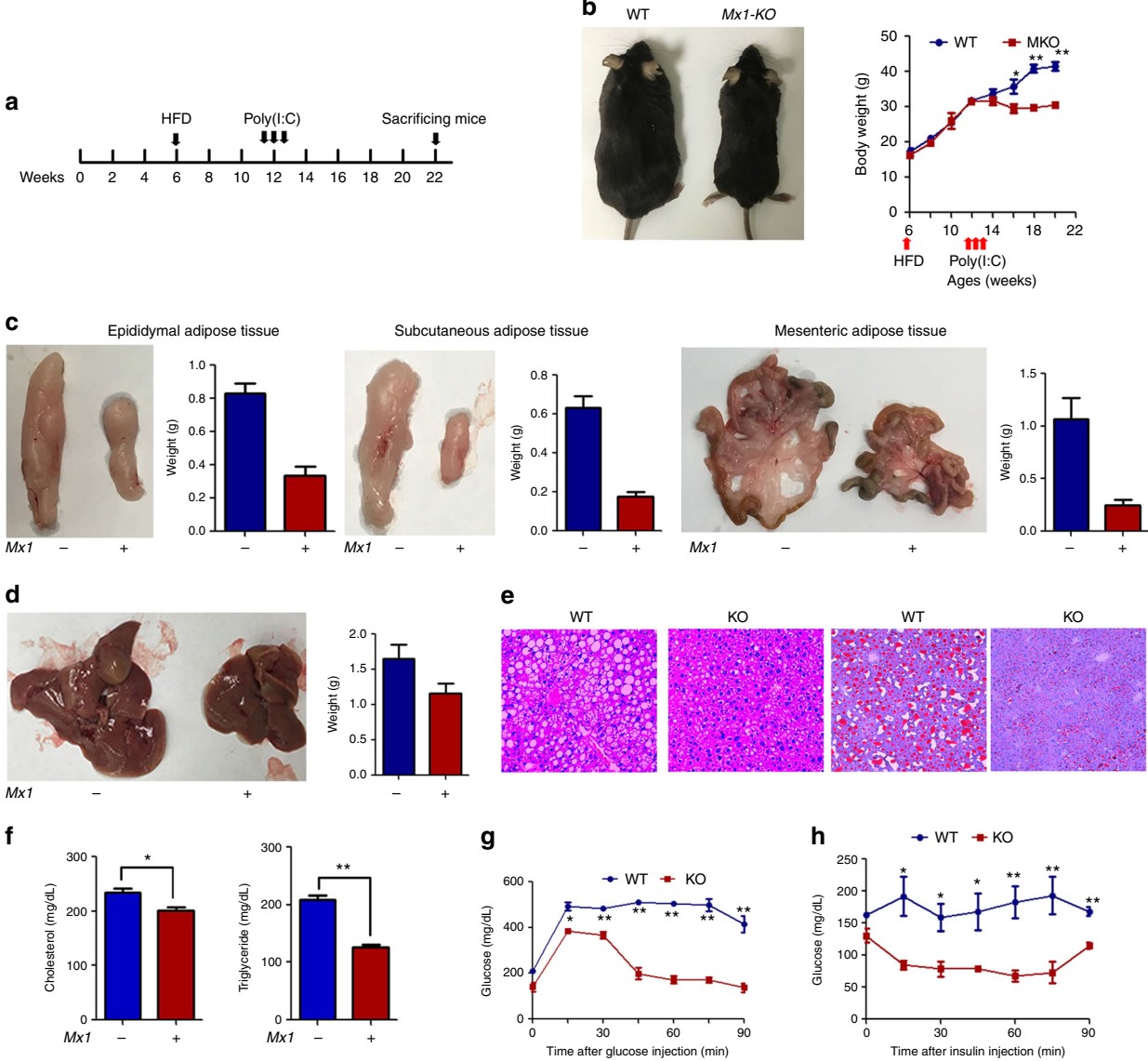

**Fig. 7** HRD1 is a potential therapeutic target for HFD-induced obesity and fatty liver disease. **a** Flowchart of the study design. **b** Body weights of male mice were measured every 2 weeks while on a HFD. Fat (**c**) and liver (*n* = 5 for each group) (**d**) weights from WT and polyI:C-injected *HRD1^F/F^*Mx1-Cre⁺ mice fed a HFD for 16 weeks. (*n* = 5 for each group). **e** H&E and Oil Red O staining of the liver from **c**. **f** Serum TG and cholesterol levels from WT and polyI:C-injected *HRD1^F/F^*Mx1-Cre⁺ mice fed with 16 weeks HFD under the refed condition. Glucose tolerance assay (*n* = 5 for each group) (**g**) and insulin tolerance assay (*n* = 5 for each group) (**h**) were performed on mice after 16 weeks of HFD. Data are representative of three independent experiments (mean ± s.d.). *P < 0.05. **P < 0.01 by unpaired Student's *t* test

metabolic disorders (Fig. 7a). Interestingly, within a week of polyI:C treatment, body weight gain in HFD-fed *HRD1^f/f^*Mx1-Cre mice fully stopped, and 3–4 weeks later, the body weight declined, despite continuation of the HFD. In contrast, control mice continued to gain body weight (Fig. 7b). Consistent with the effect on weight, we also observed that the accumulation of subcutaneous, epididymal, and mesenteric adipose tissue mass was dramatically reduced in the polyI:C-treated *HRD1^f/f^*Mx1-Cre mice (Fig. 7c). Morphological evaluation showed a pale color appearance of the liver in WT mice compared to *HRD1^f/f^*Mx1-Cre mice 10 weeks after polyI:C treatment, and both the liver size and mass of the polyI:C-treated *HRD1^f/f^*Mx1-Cre mice were significantly decreased compared to control mice, implying a reduction in lipid deposition in the liver of mice by induced HRD1 deletion (Fig. 7d). In fact, further histological

analysis demonstrated that the liver of WT control but not polyI:C-treated *HRD1^f/f^*Mx1-Cre mice developed severe steatosis 14 weeks after HFD (Fig. 7e). Staining with Oil Red O further confirmed a massive lipid deposition in the liver of control mice, which was largely diminished in the liver of *HRD1^f/f^*Mx1-Cre mice by polyI:C treatment (Fig. 6e). These results clearly indicate that liver-specific HRD1 suppression reversed HFD-induced obesity and hepatic steatosis in *HRD1^f/f^*Mx1-Cre mice. To further support this conclusion, we detected a significant reduction in the serum TG and cholesterol levels in polyI:C-treated *HRD1^f/f^*Mx1-Cre mice compared to untreated controls (Fig. 7f). In addition, polyI:C treatment significantly improved glucose tolerance and insulin sensitivity compared to WT mice (Fig. 7g, h). These results indicate that HRD1 suppression even after 6 weeks HFD

feeding fully protected mice from developing obesity, hepatic steatosis, and insulin resistance, suggesting that HRD1 is a potential therapeutic target for HFD-induced metabolic disorders.

## Discussion

The current study identified HRD1 as a novel molecular regulator in liver metabolism and a potential therapeutic target for the treatment of HFD-induced obesity and fatty liver disease. Our results from both the proteomic analysis and genome-wide RNA-seq indicate that HRD1 is a critical regulator for liver metabolic regulation including glucose, lipid, nucleotide, and amino acid metabolism. As an E3 ligase, it is not surprising that the regulatory function of HRD1 occurs through ubiquitination-mediated protein destruction. Out of the four confirmed HRD1 interactors, ENTPD5, CPT2, RMND1, and HSD17B4 were validated as HRD1 substrates, and HRD1 deletion resulted in dramatic increases in their protein, but not mRNA, levels in the mouse liver. As an ectonucleoside triphosphate diphosphohydrolase, the major effect of ENTPD5 on glycolysis might be its ability to generate ADP/AMP[27]. Elevated AMP levels (and to a lesser extent, ADP) activate AMPKs and phosphofructokinase and inhibit fructose diphosphatase to drive glycolysis and prevent gluconeogenesis and lipogenesis through downregulation of downstream target genes, including *Srebf1* and *Chrebp1*[7]. Therefore, HRD1-mediated ENTPD5 ubiquitination and degradation appears to be a critical pathway in liver metabolic regulation. Both HSD17B4 and CPT2 are multifunctional enzymes important for FA oxidation[28,29]. Therefore, the elevated activity of ENTPD5, CPT2, RMND1, and HSD17B4 caused by HRD1 deletion in the liver would facilitate FA oxidation and inhibit lipogenesis. In addition, AMPK has been reported to directly regulating the energy expenditure by modulating NAD+ metabolism and SIRT1 activity. FGF21 is a critical hepatic hormone in regulating energy homeostasis through directly targeting adipose tissue and hypothalamus. Our RNA-seq data also showed that hepatokine Fgf21 mRNA expression was dramatically increased after HRD1 deletion. Indeed, further Fgf21 deletion in HRD1-null liver reversed the increased energy expenditure induced by hepatic HRD1 deletion. However, HRD1 deletion protects mice from fatty liver disease largely in an Fgf21-independent manner. Therefore, the hepatic HRD1 appears to control mouse metabolism at least through two distinct molecular pathways: AMPK and Fgf21. A recent study has shown that HRD1 regulates PGC-1β protein stability in adipose tissues[38]; however, we observed a reduction in the expression of several PGC-1β target genes in the HRD1-null liver tissues. Therefore, HRD1 appears to function as metabolic regulator through multiple molecular mechanisms and, more importantly, in an organ-specific manner.

We found that the HRD1 interactome composes of a group of proteins in ER stress pathway, thereby confirming its function in regulating the hepatocyte unfolded protein response. HRD1-mediated degradation of misfolded proteins is critical for protecting cells from ER stress-induced apoptosis[39,40]; however, a complete deletion of HRD1 did not result in an extensive ER stress response, and both liver morphology and mass were not altered by complete HRD1 deletion. This may be due to compensatory activity of other E3 ubiquitin ligases in the ER that can reconstitute HRD1 functions in liver tissues, as several RING-finger E3 ubiquitin ligases have been identified in ER and some of them are involved in degrading misfolded proteins[38]. HRD1 has also been identified as the E3 ligase of IRE1α[14,17]. Consistent with this, we detected a significant increase in IRE1α in HRD1-null liver. Elevated activation of IRE1α has been shown to impair

insulin signaling through c-Jun N-terminal kinase[41]. In contrast, we observed a dramatic decrease in lipid and glucose levels with HRD1 deletion, implying that HRD1 regulates liver metabolism independent of the increase in IRE1α expression. In addition, while the expression of PERK-ATF4 downstream genes are elevated, treatment of Hrd1-null hepatocytes PERK-specific inhibitor did not reverse the metabolic gene expression profiles. Nevertheless, given the important roles of PERK-ATF4 ER stress pathway in metabolic regulation[37], further in vivo studies are needed to address the role of this pathway in HRD1-mediated metabolic regulation.

HRD1 appears to be a potential therapeutic target for HFD-induced obesity and fatty liver disease, because the induced deletion of HRD1 genes even after 6 weeks of a HFD prevented further body weight gain, largely diminished evidence of fatty liver disease, and abolished insulin resistance in mice. HRD1-specific inhibitors have been recently identified[24], and it will be interesting to further evaluate the therapeutic efficacy of HRD1-specific inhibitors for prevention or treatment of HFD-induced metabolic disorders. Nevertheless, given the fact that the tissue and/or organ-specific HRD1 functions are largely unstudied, it is premature to conclude that a systemic HRD1 suppression by inhibitors is an ideal therapy to treat metabolic diseases. Moreover, as a critical metabolic regulator, it is possible that HRD1 expression and/or its genetic alteration are associated with human metabolic disorders, which is under investigation by our laboratory.

## Methods

**Mice and metabolic measurements.** HRD1 floxed mice were used as previously described[20]. Hrd1-targeting vector was generated as in supplementary Fig. 1a and then transfected into an embryonic stem cell line generated from C57BL/6 mice. Neomycin selects were screened by PCR. Seven clones were obtained and confirmed by Southern blotting. Blastocyst injections resulted in several chimeric mice with the capacity for germline transmission. Breeding of heterozygous mice yielded Hrd1$^{flox/flox}$ mice without phenotypic abnormalities in expected Mendelian ratios. Mx1-Cre (catalog no. 003556) and Albumin-Cre (catalog no. 016832) mice are at the C57BL/6 genetic background and purchased from The Jackson Laboratory. All mice were bred in a specific pathogen-free facility, and all animal experiments were approved by the institutional animal care and use committees at the Northwestern University. Animals were maintained on a standard chow diet under 12-h light and dark cycles beginning at 5:00 a.m. and 5:00 p.m., respectively. No randomization and blinding were used for the animal study. HFD (45 kcal% fat) is from the Research Diet Inc.

Measurements of food intake, activity, $O_2$ consumption, $CO_2$ production, and RER with indirect calorimetry were performed using a comprehensive laboratory animal monitoring system.

**Plasmids and antibodies.** The full-length and truncation mutants of HRD1 and the ubiquitin expression plasmid were constructed as reported previously[19,20]. Mouse HSD17B4, ENTPD5, ATP5D, and CPT2 gene were amplified by PCR from mouse liver complementary DNA (cDNA) and cloned into pCMV-Myc (Clontech, CA, USA). The human GFP-SHP1 plasmid was obtained from Addgene (Cambridge, MA, USA). Antibodies to Flag (F1804), HRD1 (HPA024300), HMGCR (SAB4200529), and ATP5D (HPA002865) were purchased from Sigma (St. Louis, MO, USA), and antibodies to Myc (9E10), Myc (9E10)-HRP and HA (F7)-HRP were purchased from Santa Cruz (Santa Cruz, CA, USA). Anti-ENTPD5 (ab108603), anti-RMND1 (ab98850), anti-HSD17B4 (ab170910), and anti-MLYCD (ab95945) were purchased from Abcam (Cambridge, MA, USA). All other antibodies including Actin (13E5), AMPK and ACC Antibody Kit, GAPDH (14C10) pan-AKT (4691), p-AKT (2965), p-AKT (2965), S6K (2708), p-S6K (2215), Foxo1 (2880), and p-Foxo1 (9464) were purchased from Cell Signaling Technology (Danvers, MA, USA). All antibody working concentration for immunoblotting is 1 µg/ml.

**Cell culture, transfection, and immunoprecipitation.** HEK293 cells (ATCC CRL-11268G-1) and HepG2 (ATCC HB-8065) cells were cultured in Modified Eagle's medium (MEM) supplemented with 10% fetal bovine serum. Transfections were performed with Lipofectamine 2000 (Invitrogen, Carlsbad, CA) according to the manufacturer's instructions. HEK293 cells were transfected with the indicated expression vectors. After 24 h, the cells were lysed in lysis buffer supplemented with protease (Roche, Basel, Switzerland) and phosphatase inhibitor cocktails. Co-IP and immunoblotting analyses were performed as described previously[42,43]. Cells

were lysed in cold radio immunoprecipitation assay buffer (EMD Millipore catalog no. 20-188). The cell lysate was precleaned with protein-G sepharose (GE Healthcare, catalog no. 17-0618-02) for 30 min and subjected to immunoprecipitation with each indicated antibody (2 μg), incubated for 1 h on ice, and then 25 μl of protein-G sepharose beads were added for additional 4–6 h. The beads were then washed for four times, boiled with 15 ml of 2 × sodium dodecyl sulfate (SDS) sample buffer for 5 min and the proteins were separated on 8–10% SDS–polyacrylamide gel electrophoresis (PAGE) gels and transferred to polyvinylidene difluoride membranes. The membranes were blocked in 5% fat-free dried milk in Tris-buffered saline with 0.5% Tween 20 (TBST) for 2 h. The membranes were then incubated in appropriate primary antibodies overnight at 4 °C. Membranes were washed in TBST and then incubated in horseradish peroxidase (HRP)-conjugated secondary antibodies (1:2500) (EMD Millipore Corp, goat anti-rabbit IgG antibody, HRP conjugate, catalog no. 12–348; goat anti-mouse IgG antibody, HRP conjugate, catalog no. 12–349). Membranes were washed in TBST, and the signals were visualized using enhanced chemiluminescence substrate (Thermo Scientific, Waltham, MA) and quantified using the Bio-Rad Image software. Representative uncropped blots are shown in Supplementary Fig. 15, 16.

**AP-MS analysis**. HepG2 cells were cultured in MEM supplemented with 10% (v/v) fetal bovine serum and MEM Non-Essential Amino Acids Solution (HyClone, Logan, UT, USA). Cultures were grown to approximately $1 \times 10^7$ cells. Briefly, cells were incubated on ice for 30 min in NETN lysis buffer (20 mM Tris-Cl [pH 8.0], 150 mM NaCl, 1 mM EDTA, 0.5% NP-40) supplemented with freshly added completely EDTA-free Protease Inhibitor (Roche), 0.1% 1 M dithiothreitol, and 1 M $Na_3VO_4$. The lysates then underwent ultracentrifugation (100,000 r.c.f.) for 20 min at 4 °C. The supernatants were collected and incubated with normal rabbit IgG or HRD1 (SYVN1) antibodies (Cell Signaling) for 2 h followed by ultracentrifugation and 1.5 h incubation with Protein A Sepharose™ CL-4B (GE Healthcare, Sverige, Sweden). The beads were washed with NETN lysis buffer and the binding proteins were eluted. Lysates and immunoprecipitates were detected using the indicated primary antibodies, followed by detection with ECL substrate. The protein samples were resolved by 10% SDS-PAGE and stained with Coomassie Brilliant Blue R250 in parallel. The full lanes were cut into small bands, regardless of the presence of visible bands. The bands were digested with 100 ng of trypsin overnight, extracted twice with 100% acetonitrile, and dried in a Savant Speed-Vac. The peptides were resolved in 0.1% formic acid and were analyzed using an ultra-performance liquid chromatography (LC)–tandem mass spectrometry platform. The LC separation was performed on an Easy nLC 1000 (Thermo Fisher Scientific, San Jose, CA, USA) with an in-house packed capillary column (150 μm I.D. × 12 cm) with 1.9 μm C18 reverse-phase fused-silica (Michrom Bioresources, Inc., Auburn, CA, USA). The sample was eluted with a 78 min nonlinear gradient ramped from 5% to 30% mobile phase B (phase A: 0.1% formic acid in water, phase B: 0.1% formic acid in acetonitrile) at a 0.6 μl/min flow rate. Eluted peptides were analyzed by a Q-Exactive orbitrap mass spectrometer (Thermo Fisher Scientific, San Jose, CA, USA). The MS1 was analyzed over a mass range of 300–1400 Da with a resolution of 70,000 at $m/z$ 200. The isolation width was 3 $m/z$ for precursor ion selection. The automatic gain control (AGC) was set to $3 \times 10^6$, and the maximum injection time (MIT) was 60 ms. The normalized collision energy was 27 ev. Top 20 MS2 spectra were acquired with a resolution of 17,500 at $m/z$ 200; the AGC was set at $5 \times 10^4$ and the MIT was 80 ms. The dynamic exclusion was set at 18 s. Spectral data were then searched against Uniprot Swiss-Prot Human (v201503) database with the Mascot v2.3.2 software. The peptide and protein false discovery rates were set to 0.01. The cutoff ion score from Mascot was set to 10. SILVER software was used to get the spectral counts of each protein[44]. Owing to the fact that no contaminant data were available for HepG2 cells, the common contaminants were filtered by cRAP protein sequences from gpmDB [http://www.thegpm.org/crap/index.html] and contaminants from MaxQuant [http://www.coxdocs.org/doku.php?id=maxquant:start]. The COMPASS and SAINT scoring methods were used to rank the HRD1-interacting proteins[45]. The raw MS data were deposited in iProx database with identifier IPX0000928000 [http://www.iprox.org/index].

**Genome-wide RNA-seq and real-time qPCR**. WT and HRD1ALB mice were fasted overnight and refed for 4 h. For the genome-wide RNA-seq, total hepatic RNA was isolated with TRIzol reagent and cDNA fragments were prepared for Illumina NextSeq 500. Two samples were prepared for each condition, and each sample was run on one lane for sequencing, with the number (proportion) of successfully aligned reads about 50 million (deposited in GEO [https://www.ncbi.nlm.nih.gov/geo/]). Counts of reads and RPKM (reads per kilo-base pair per million reads mapped) values for each gene were determined using the ERANGE software and were tested for differential expression in R using the limma package and the IBMT (intensity-based moderated $t$ statistic) method.

cDNA was synthesized using the qScript cDNA Synthesis Kit (Quanta Biosciences, Gaithersburg, MD, catalog no. 95047-100). iQ5 and SYBRGreen Detection system (Bio-Rad, Hercules, CA) were used for qPCR. qPCR was run on a Bio-Rad IQ2 PCR machine, and each PCR mixture contained 40 ng of cDNA template and 10 nM primers in 15 μl of SYBR green reaction mix (Bio-Rad). The Expression values were normalized to those that were obtained with the control *actb* (encoding beta-actin). All the $R^2$ (coefficient of determination) of the reaction

with specific primers is >0.99 and the efficiency of the PCR reaction in the range 100% ± 10% is as shown in the protocol from Thermo Fisher Scientific. Changes in gene expression levels were calculated by the $2^{-\Delta\Delta Ct}$ method[43].

**Histological analysis**. H&E staining was performed as shown in a published protocol[43]. Briefly, liver tissues from mice were fixed overnight in 10% neutral formalin and embedded in paraffin. Paraffin-embedded tissues were cut into sections and stained with H&E. For the Oil Red O staining, liver tissues from mice were embedded in Tissue-Tek OCT in a cryostat mold and cut into 10 μm sections. Tissue sections were stained with 0.5% Oil Red O and counterstained with Mayer's hematoxylin.

**Blood chemistry analysis**. Secreted mouse insulin in serum were determined using the commercially available ELISA kits (R&D, USA), followed by analysis with Multiskan MK3 (Thermo Scientific). Blood glucose, serum cholesterol, and TG levels were determined according to the manufacturer's instructions (Abcam, USA).

**Statistical analyses**. Statistical analysis was performed using Student's $t$ test, and $P < 0.05$ was considered significant. Data are expressed as mean ± SEM.

## Data availability

Sequence data that support the findings of this study have been deposited in "GenBank" with the [primary] accession code "KP253039". The raw MS data were deposited in iProx database with identifier IPX0000928000. All data are available from the authors upon reasonable request.

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

## Acknowledgements

This work was supported by grants from the Special Funds for Major State Basic Research of China (2014CBA02001) to F.H. and J.W. and the National Institutes of Health (NIH) R01 grants (AI079056, AI108634, and AR006634) to D.F., NIH grant DK090313 to K.Z., and NIH grant 5P01AG049665 to N.S.C.

## Author contributions

J.W., Y.Yuan, L.C., Y.W., Y.Z., Y.X., C.B.P., L.D., Y.Yang, B.G., J.M.-C., and C.J. performed the experiments and analyzed the data. Y.Yang, B.G., J.M.-C., D.D.Z., and K.Z. contributed critical reagent and in experimental design. H.P., F.H., N.S.C., J.W., and D.F. designed the study, analyzed the data, and wrote the manuscript.

## Additional information

**Competing interests:** The authors declare no competing interests.

