## [Peer Review File · Nature Communications]

Editorial Note: Figure 6e was removed from the manuscript post acceptance since the authors alerted the editors to an error in the way these results were calculated. The main conclusions of the manuscript were not altered, but the results and discussion sections were revised to reflect this change and the panels in Figure 6 were renumbered accordingly.

Reviewers' comments:

Reviewer #1 (Remarks to the Author):

Review of Wei et al, "Hepatic ER-associated ubiquitin ligase HRD1 is a therapeutic target for high-fat diet-induced metabolic syndrome," Nature Communications

In this manuscript, the authors describe the phenotype of a liver-specific knockout of the ERAD-associated E3 ubiquitin ligase HRD1. Extensive and convincing data support that Hrd1 LKO animals are protected from metabolic syndrome-like effects, including the metabolic dysregulation that accompanies high fat diet. Importantly, this effect can even be induced after the fact by induced deletion of Hrd1. In order to account for the phenotype, the authors then show that several metabolic proteins are likely targets of HRD1 ligase activity, and that loss of Hrd1 also results in extensive changes in the expression of metabolic genes.

By and large, the data in this paper are of high quality, and the finding that deletion of HRD1 in the liver results in increased energy expenditure and apparent tissue non-autonomous effects in other organs (including adipose) is intriguing and of interest. However, some of the major claims of the paper need to be tempered considerably:

1. The authors claim that degradation of ENTPD5, CPT2, and HSD17B4 is ultimately responsible for activation of AMPK and AKT and the consequent metabolic alterations. This is not shown directly, and is not tested directly; the apparent degradation of these substrates and activation of AMPK and AKT (and the changes in gene expression) might be entirely coincidental.
2. The authors claim that ER stress is not induced by HRD1 deletion and is thus not responsible for the phenotype. Based on their array data (Table S3), I disagree. In addition to Ddit3, Trib3, and Gadd45, many other canonical UPR genes are also upregulated by Hrd1 knockout, including BiP (Hspa5—the most common marker of UPR activation), Grp94 (Hsp90b1), Derl3, Xbp1, Pdia6, Pdia3, and others. Thus, even though there is no obvious histological effect of Hrd1 deletion, it seems quite clear that ER stress is being elicited (and this effect might even be more obvious in acute deletion of Hrd1, such as by Poly I:C or Ad-CRE). This might (or might not) account for some (or all) of the observed phenotype—indeed, the fact that, paradoxically, both lipogenic and fatty acid oxidation genes (at least including Ppara in the latter group) are suppressed is strongly reminiscent of the effects of acute ER stress in the liver (except in that case the liver becomes more steatotic, rather than less). The authors might in fact have arrived upon a viable model for chronic genetic ER stress in the liver.
3. The authors claim that glycogenesis and lipogenesis are suppressed but glycolysis and FA oxidation are enhanced. This cannot be concluded because the activities of these pathways are not directly tested (indeed, the fact that Ppara is suppressed might suggest that FA oxidation is also suppressed; in any case, there is no systematic examination of even FA oxidation genes, let alone pathway activity).
4. It is not clear, ultimately, why Hrd1 LKO animals have an elevated metabolic rate (and it is a bit surprising that both AMPK and AKT are activated, as these pathways generally act at cross-purposes; in fact, the authors state that the AKT pathways suppresses lipogenesis, but actually the converse is true—AKT activates lipogenesis; see PMID 21723501 and others). The authors need to be clear that their data only show that loss of Hrd1 leads to systemic alterations to metabolism, but that the mechanism by which these lead to reduced weight gain, elevated O₂ consumption, diminished adipose stores, etc., is not known.

The amount of work required to address each of these issues would be substantial and, more to the point, beyond the essential scope of this paper given its strengths. However, extensive careful rewriting to take these considerations into account is required.

Other points:

1. Were mice of both genders used? This is not stipulated anywhere that I can find. If only one

gender, this needs to be made clear (and ideally at least one or two key experiments need to be repeated in the opposite gender and the effects if any, shown).

2. Each figure panel should be clear about the number of animals used in each experiment. Also, each legend should include more information about animal ages, fed versus fasted state, etc. for all experiments.

3. How did the authors verify that their qRT-PCR primers are (a) specific, (b) quantitative, and (c) linear across a wide dynamic range? This is not stipulated, and if the primers were not piloted by standard curving, then the qRT-PCR data are not trustworthy.

4. I do not agree that HSD17B4 is an HRD1 substrate. While its steady state expression increases when Hrd1 is deleted, the ubiquitylation of HSD17B4 (Fig. 4B, C) does not to my eye look significantly different.

5. Gels showing apparent increased phosphor-AMPK α and phosphor-ACC (Fig. 4D) are not particularly convincing, which makes the provided quantification also not particularly convincing. Authors should show the gels from more than two animals in each group.

Reviewer #2 (Remarks to the Author):

This interesting manuscript by Wei and Yuan et al., suggests that liver specific HRD1 deficiency results in increased energy expenditure and lower body weight, blood glucose and plasma lipids. Proteomic analysis identified HRD1 to interact with several metabolic enzymes including ENTPD5, CPT2, RMND1 and HSD17B4. Loss of hepatic HRD1 results in hyperactivation of AMPK and AKT signaling pathways and facilitates glycolysis and fatty acid oxidation. Although the results are intriguing, there are some issues that require further attention as detailed below.

Major:

1- Overall, the phenotype of HRD1 LKO mice is increased energy expenditure and lower body weight which –perhaps– explains the lower blood glucose and plasma lipids. However, no insight into how hepatocyte-specific deletion of a gene results in increased energy expenditure is provided. The authors show many interesting findings in liver after HRD1 is deleted (i.e. increased AMPK and AKT activity, etc.) but none of them explains why these mice have increased energy expenditure. It would be nice to speculate a mechanism. For example, is the reason why these LKO mice have increased energy expenditure because they have increased beigeing/browning which may suggest a liver-adipose tissue crosstalk?

2- The results presented in Figure 4 and 5 showing that HRD1 deficiency results in activation of AMPK and AKT signaling and lower expression of genes involved in fatty acid and TG synthesis. Are these effects cell intrinsic to hepatocytes? What's the evidence that these changes are NOT due to lower body weight in LKO mice? Similar analysis after acute silencing of HRD1 using primary hepatocytes or hepatocyte cell lines should be done in order to suggest that these results are primarily due to HRD1 deficiency.

3- Do hepatic HRD1 levels change upon high-fat diet feeding or other models (genetic) of obesity?

Minor:

1- Blood glucose is only regulated after refeeding –how about plasma insulin levels? Do HRD1 LKO mice have lower refed blood glucose levels because they secrete more insulin after feeding?

2- Were TG levels in Figure 1e measured in fasting or refeeding? Do TG levels change after refeeding only -like blood glucose?

3- Although the authors provide H&E staining of liver and show no abnormality, liver enzymes (ALT and AST) should be measured to prove that HRD1 deficiency does not result in liver toxicity.

4- The degree of hepatic HRD1 silencing should be shown in Hrd1^{fl/fl} Mx1-Cre mice.

5- Blood glucose levels in Fig s3a, Fig 2f, Fig 2g are expressed as mM –I believe those are mg/dl.

6- Adenovirus-mediated gene delivery is transient and usually lasts 10-14 days. It is interesting to see that adeno-Cre injected mice had less body weight gain even after 1 month of adeno-Cre treatment. It would be nice to provide the liver HRD1 levels to show that the adeno-Cre effect,

namely silencing of HRD1, is still present 1 month after adenovirus injection.

7- Figure 2g and 6h are described as insulin tolerance tests in figure legends and written as "glucose infusion rate upon hyperinsulinemic clamping" in the text. The text should be changed to "insulin tolerance test" not to confuse insulin tolerance test with hyperinsulinemic euglycemic clamp studies which is completely different and was not performed in this study.

8- Figure 2i is missing significance asterisk/signs.

9- Figure 4c blots are too dark and hard to interpret. It would be nice to see lower exposed western blot data to clearly argue that ubiquitination is decreased.

10- Figure 4d is missing the actin blots. The densitometric quantification is done with actin and not shown in the figures.

Reviewer #3 (Remarks to the Author):

Identifying environmental and genetic factors that disrupt metabolic processes in obesity and chronic over-nutrition are a critical avenue of scientific inquiry for addressing metabolic disease clusters in modern society. Despite much progress, basic features driving pathogenesis remain unclear. Here, Wei & Yuan et al consider the broad contribution of dysregulated catalytic enzyme activity in the collective concept of metabolic syndrome, focusing on the relatively unexplored area of protein degradation/turnover. This inquiry leads the authors to a detailed investigation of HRD1, an E3 ligase known to play a role in targeting degradation of misfolded proteins in the ER by the process of ERAD. Based on this foreknowledge, one might have expected genetic disruption of HRD1 in livers of high fat diet fed mice, which are known to exhibit elevated ER stress, would have caused severe problems in liver metabolism. Quite the contrary however, Wei & Yuan et al show that these animals display several phenotypic improvements including higher energy expenditure, reduced weight gain & adiposity, and protection against hepatic steatosis as well as better lipid profiles, insulin sensitivity, and glucose homeostasis.

The authors further show using unbiased proteomic analysis that the phenotype observed in the HRD1 deficient mice seems to be less related to ERAD but instead through a previously unrecognized role of HRD1 in regulating turnover rates of key metabolic enzymes ENTPD5, CPT2, RMND1, and HDS17B4, as well as some other expected targets. Overall, this study has several novel and powerful elements compelling enough to support its publication. That being said, I have a few issues that I think are worth consideration.

Issues:

1) The overall study implies that nutrients regulate HRD1 in the liver following feeding, causing HRD1 to cause degradation of these enzymes. As such, deletion of HRD1 causes the enzymes to be expressed much higher. However, the authors really only look at the role of fasting/feeding on HRD1 levels (and not its actual activity) in one panel (Figure 1b). I think this should be expounded upon. Specifically, it would be highly desirable that the authors correlate the feeding induced rise in HRD1 levels with the levels of at least one of the enzymes mentioned (ENTPD5, CPT2, RMND1 or HDS17B4). Whether the activity responds to feeding and regulation is lost in HRD1 deficiency is of interest and would add to the conclusions.

2) Along these lines, the authors state "we found that HRD1 was postprandially induced to control the metabolic balance in the physiological condition." However, the authors do not discuss the role of such a regulatory circuit in a physiological setting. Specifically, how might and for what function would HRD1 target these enzymes following a meal?

3) While the phenotype of the HRD1 deficient model is striking, the energy expenditure result is almost bewildering. Can the authors explore how a change in liver metabolism has such a profound effect on whole body energy expenditure? Any insight into this would be very interesting

and at the very least the discussion should be expanded.

4) On line 160 and 161 of the text it describes the use of hyperinsulinemic clamps in figure 2f. However, figure 2f shows a glucose tolerance test, not a clamp study. Please correct.

5) Detailed information should be provided in the generation and housing of the control and HRD1-deficient mice, the nature of the diets, and their genetic background.

Reviewer #4 (Remarks to the Author):

The manuscript by Wei et al. describes the detailed analysis of HRD1 function in the liver of mice. While the protein was described as a regulator of ER stress, the authors here show a link of this ubiquitin ligase with various aspects of metabolism. They reveal an important role for the protein in energy expenditure and demonstrate that loss of the protein results in resistance against high-fat diet-induced obesity and liver steatosis. A conditional mouse model supports the targeting of this protein as a therapeutic protein.

Experiments are well performed and overall the data looks convincing and of high quality. The manuscript is well written and has a logical structure.

I only have a few minor comments on the manuscript:

I think that the current title of the manuscript ('Hepatic ER-associated ubiquitin ligase HRD1 is a therapeutic target for high-fat diet-induced metabolic syndrome') should be less focused on the therapeutic aspect of the HRD1 protein. While there is some support that HRD1 can be a good therapeutic target, there is still some additional data needed to confirm this. Moreover, the authors do not really discuss how such a therapeutic targeting of HRD1 would be realized.

For the proteomics experiments it would be good if the authors clearly describe how many controls were actually performed. From the SC numbers I am assuming that 2 controls were performed. More controls would have been better. Since SC numbers are markedly different and since the authors provide sufficient validation for the selected candidates the data is overall very convincing. Combined use of COMPASS and SAINT is a solid strategy to prioritize candidates. It is not fully clear what drove the selection of the candidates for further validation. Were additional criteria used or were these proteins used to represent certain biological processes? Did the authors use the CRAPome for filtering?

The proteomics data should be uploaded to a repository (e.g. PRIDE) so it is available for other scientists.

RT-qPCR experiments are more reliable when more housekeeping genes are used.

Fig 3g. Please specify clearly if the detection of the candidate interactors is with an epitope tag (overexpression) or with specific antibodies.

Full gel views should be made available in supplementary information.

Some typos and other suggested improvements:

Abstract line 22: please start with the full name of the protein

Line 29: Genome-wide mRNA sequencing

Line 105: Similarly, we detected dramatically lower...

Fig S6: resolution currently not sufficient. Integration of the provided figure is required.

We thank the reviewers for their interest and constructive suggestions, which lead to significant improvement to our study. We have carefully examined the critiques and addressed them below as well as in the revised proposal. Changes made to the proposal are underlined.

Reviewer #1 (Remarks to the Author):

1. The authors claim that degradation of ENTPD5, CPT2, and HSD17B4 is ultimately responsible for activation of AMPK and AKT and the consequent metabolic alterations. This is not shown directly, and is not tested directly; the apparent degradation of these substrates and activation of AMPK and AKT (and the changes in gene expression) might be entirely coincidental.

Reply: To further validate our initial conclusion that HRD1-deficiency affects liver metabolism through AMPK pathway, we asked whether the AMPK-specific inhibitor (compound C) reverse the metabolic alteration induced by HRD1 deletion. Indeed, the expression of genes involved in lipogenesis including *Chrebp1*, *Scd1*, *Gyk* was significantly increased by AMPK inhibitor injection in HRD1 LKO mice. In addition, up-regulation of the genes regulating the fatty acid oxidation (*Lpl* and *Acot9*) in HRD1 LKO mice was reversed by AMPK inhibitor administration (Figure 6d & s11). Together with our data that AMPK phosphorylation is dramatically increased in HRD1 KO liver cells (Figure 4d), these results indicated that metabolic alteration induced by HRD1 gene deletion is due to, at least partially the AMPK over-activation.

Figure 6d & s11. Effect of AMPK inhibitor on the gene expression in HRD1-null hepatocyte. WT and HRD1 LKO mice were fasted overnight and refed for 1 hour and then administrated with AMPK inhibitor for additional 3 hours. Hepatic mRNA level of *Chrebp1*, *Scd1*, *Gyk*, *Srebf1*, *Lpl* and *Acot9* were measured.

We discovered that HRD1 functions as an E3 ubiquitin ligase of ENTPD5, an enzyme known to activate AMPK. To further determine whether the liver-specific HRD1 deletion alters liver metabolic response through the elevated ENTPD5, we infected primary hepatocytes from WT and HRD1 KO mice with lentivirus expressing *Entpd5* specific shRNA. As indicated in Figure S14, *Entpd5* specific shRNA efficiently knocked down ENTPD5 mRNA without altering *Hrd1* mRNA expression. Importantly, suppression of ENPTD5 expression partially rescued *Chrebp1* and *Scd1* expression in HRD1-null hepatocytes. In addition, the elevated *Lpl* expression in HRD1-null liver cells was largely reversed by *Entpd5* knockdown. These results indicated that metabolic alteration induced by HRD1

deletion partially through the ENTPD5 accumulation.

Figure 6f. Effects of ENTPD5 knockdown on metabolic gene expression in HRD1-null liver cells. WT and HRD1 LKO hepatocytes were infected with lentivirus to specific knockdown *Entpd5*. 2 days after infection, Hepatic mRNA level of *Entpd5*, *Hrd1*, *Chrebp1*, *Gyk*, *Lpl* and *Srebf1* were measured.

2. The authors claim that ER stress is not induced by HRD1 deletion and is thus not responsible for the phenotype. Based on their array data (Table S3), I disagree. In addition to *Ddit3*, *Trib3*, and *Gadd45*, many other canonical UPR genes are also upregulated by *Hrd1* knockout, including BiP (*Hspa5*—the most common marker of UPR activation), *Grp94* (*Hsp90b1*), *Derl3*, *Xbp1*, *Pdia6*, *Pdia3*, and others. Thus, even though there is no obvious histological effect of *Hrd1* deletion, it seems quite clear that ER stress is being elicited (and this effect might even be more obvious in acute deletion of *Hrd1*, such as by Poly I:C or Ad-CRE). This might (or might not) account for some (or all) of the observed phenotype—indeed, the fact that, paradoxically, both lipogenic and fatty acid oxidation genes (at least including *Ppara* in the latter group) are suppressed is strongly reminiscent of the effects of acute ER stress in the liver (except in that case the liver becomes more steatotic, rather than less). The authors might in fact have arrived upon a viable model for chronic genetic ER stress in the liver.

Reply: We appreciate this great suggestion. Indeed, it has been well-established that the ER stress is important for the hepatic lipogenesis. Our data that the PERK-ATF4 pathway target gene *Ddit3* (*Chop*) but not IRE1 α downstream gene *Xbp1s* was increased after HRD1 deletion, implying a possibility that HRD1-deficiency alters liver metabolism through PERK-ATF4 activation. To test this possibility, we injected a PERK-specific inhibitor to HRD1 LKO mice and asked whether PERK inhibition reverses the gene expression profile by HRD1 deficiency. *Ddit3* mRNA expression was dramatically inhibited by PERK inhibitor, confirming that PERK-ATF4 pathway was indeed blocked. In contrast, the expression levels of *Srebf1*, *Chrebf1*, *Scd1*, *Fasd2*, *Gyk*, *Agpat2*, *Apob* and *Tymp* in liver of both WT and HRD1 KO mice were unaltered by PERK inhibitor administration (Figure S9). These results indicate that HRD1 regulate the lipid metabolism in an ER stress independent manner.

Figure S9. The effects of PERK inhibitor on metabolic responsive gene expression in HRD1 KO liver. (a) Hepatic *Xbp1s* were measured from WT and HRD1 mice in a refed condition. (b-d) WT and HRD1 LKO mice were fasted overnight and refed for 1 hour and then administrated with PERK inhibitor for additional 3 hours. Hepatic mRNA level of *Ddit3 (Chop)* (b), *Srebf1* and *Chrebp1*(c) *Scd1*, *ApoB*, *Fads2* and *Tymp* (d) were measured.

3. The authors claim that glycogenesis and lipogenesis are suppressed but glycolysis and FA oxidation are enhanced. This cannot be concluded because the activities of these pathways are not directly tested (indeed, the fact that *Ppara* is suppressed might suggest that FA oxidation is also suppressed; in any case, there is no systematic examination of even FA oxidation genes, let alone pathway activity).

Reply: We agree with the reviewer. Our data imply rather than conclude “that glycogenesis and lipogenesis are suppressed but glycolysis and FA oxidation are enhanced in HRD1-null mice”. Unfortunately, the laboratory has no authorization to use radioactive reagents to directly analyze the FA oxidation and it will take time to obtain the authorization in particular for the use in animals. Nevertheless, we have edited to “our following observations suggest that HRD1 deletion enhances FA oxidation in mice”, which is supported by the following evidences: Firstly, we found that HRD1 deletion increased the mitochondrial biogenesis (126 mitochondrial gene expression increased, Table S6). Secondly, we found that the mRNA including *Acot2*, *Acot5*, *Acot9* (Fatty acid oxidation), *Fabp4*, *Fabp7*, *Slc27a3*, *Aldh18a1* (Fatty acid transportation) and *Lpl* (TG metabolism) were dramatically increased but no genes mediating the FA oxidation were decreased after HRD1 deletion (Figure S8). Thirdly, CPT2, RMND1 and HSD17B4 are the metabolic enzymes directly regulating the FA oxidation. Our data also showed that the protein levels of CPT2, RMND1 and HSD17B7 were dramatically increased after HRD1 deletion. These results indicated that FA oxidation is increased after hepatic HRD1 deletion.

Figure S8. The genes involved in TG and fatty acid oxidation were increased after HRD1 deletion. Hepatic *Acot2*, *Acot9*, *Fabp7* and *Lpl* were measured from WT and HRD1 mice in a refed condition.

4. It is not clear, ultimately, why *Hrd1* LKO animals have an elevated metabolic rate (and it is a bit surprising that both AMPK and AKT are activated, as these pathways generally act at cross-purposes; in fact, the authors state that the AKT pathways suppresses lipogenesis, but actually the converse is true—AKT activates lipogenesis; see PMID 21723501 and others). The authors need to be clear that their data only show that loss of *Hrd1* leads to systemic alterations to metabolism, but that the mechanism by which these lead to reduced weight gain, elevated O₂ consumption, diminished adipose stores, etc., is not known.

Reply: We agree with the reviewer and now clarified that loss of HRD1 functions leads to systemic alterations to metabolism, which is possibly due to, at least partially, the elevated hepatokine FGF21. This is supported by the following evidences: First we discovered that HRD1 deletion increased the *Fgf21* expression from our RNA-Seq data. Second, we confirm the elevation of *Fgf21* mRNA by qRT-PCR methods and we also found the FGF21 protein levels in the serum were increased after hepatic HRD1 deletion. Third, the elevation of energy expenditure of the HRD1 LKO mice was rescued by FGF21 deletion. However, deletion of FGF21 failed to rescue the decreased levels of serum TG and FA in HRD1 LKO mice and Stearoyl-CoA Desaturase (*Scd1*) expression (Fig. 6B) which catalyze the rate-limiting step in the formation of monounsaturated fatty acid, suggesting that HRD1 may regulate TG and FA metabolism through alternative pathways in the absence of FGF21.

Figure S10. Elevation of energy expenditure of HRD1 LKO mice through FGF21 overexpression. (a) Hepatic *Fgf21* mRNA in the WT and L-HRD1 KO mice. (b) Serum FGF21 protein levels in the WT and L-HRD1 KO mice. (c) Energy expenditure of WT, HRD1 LKO and HRD1/FGF21 DKO mice. (d) Serum fatty acid and TG levels of WT, HRD1 LKO and HRD1/FGF21 DKO mice. (e) Relative *Scd1* mRNA expression of WT, HRD1 LKO and HRD1/FGF21 DKO mice.

Other points:

1. Were mice of both genders used? This is not stipulated anywhere that I can find. If only one gender, this needs to be made clear (and ideally at least one or two key experiments need to be repeated in the opposite gender and the effects if any, shown).

Reply: Both genders were used for the studies, which is now clarified in the revised manuscript.

2 Each figure panel should be clear about the number of animals used in each experiment. Also, each legend should include more information about animal ages, fed versus fasted state, etc. for all experiments.

Reply: All the figure legends have revised as suggested.

3. How did the authors verify that their qRT-PCR primers are (a) specific, (b) quantitative, and (c) linear across a wide dynamic range? This is not stipulated, and if the primers were not piloted by standard curving, then the qRT-PCR data are not trustworthy.

Reply: Our qPCR results are consistent with the unbiased RNA-Seq screening data. Nevertheless, as now described in the Method section of our manuscript, we used the comparative CT method to measure the relative changes of relative gene expression. We assess whether our intercalating dye qPCR assays have produced single, specific products by melt curve analysis of the PCR. To follow the suggestion of the reviewer, the standard curve was constructed by plotting the log of the starting quantity of template against the CT value obtained during amplification of each dilution. The equation of the linear regression line, along with the coefficient of determination (R^2) was used to evaluate whether our qPCR assay is optimized. As showed in Table S7, All the R^2 of the reaction is higher than 0.99 and the efficiency of the PCR reaction in the range $100\% \pm 10\%$ as showed the protocol from Thermo Fisher Scientific.

	Actb	Srebp1	Chrebp 1	Scap	Scd1	Fasn	Fads2	Gyk
R^2	0.9999	0.9977	0.999	0.9971	0.9999	0.9999	0.9983	0.9989
Efficiency	0.972	1.05	0.95	1.01	1.01	1.1	1.04	1.01
	AGPAT 2	AGPAT 6	Apob	Upp2	Entpd5	RMND1	Hsd17b 4	Cpt2
R^2	0.9998	0.991	0.9989	0.9984	0.999	0.997	0.9985	0.9996
Efficiency	0.981	1.09	1.05	0.977	0.992	0.985	0.995	1.021

4. I do not agree that HSD17B4 is an HRD1 substrate. While its steady state expression increases when *Hrd1* is deleted, the ubiquitylation of HSD17B4 (Fig. 4B, C) does not to my eye look significantly different.

Reply: The samples were blotted with anti-Ub again and exposed shorter. As showed in Figure 4c, the endogenous ubiquitination level of HSD14B4 was decreased after hepatic HRD1 deletion. The figures are revised as suggested.

5. Gels showing apparent increased phosphor-AMPKa and phosphor-ACC (Fig. 4D) are not

particularly convincing, which makes the provided quantification also not particularly convincing. Authors should show the gels from more than two animals in each group.

Reply: The western blotting conditions for analyzing phosphor-AMPK α and phosphor-ACC have been optimized, and the levels of both phosphor-AMPK α and phosphor-ACC in the froze cell lysates from the same set of mice were reanalyzed. Images with better resolution are now provided (Figure 4a, panels 1 & 4)

Figure 4 (a) HRD1-binding protein levels in the liver of WT and HRD1 LKO mice under the refed condition.

Reviewer #2 (Remarks to the Author):

This interesting manuscript by Wei and Yuan et al., suggests that liver specific HRD1 deficiency results in increased energy expenditure and lower body weight, blood glucose and plasma lipids. Proteomic analysis identified HRD1 to interact with several metabolic enzymes including ENTPD5, CPT2, RMND1 and HDS17B4. Loss of hepatic HRD1 results in hyperactivation of AMPK and AKT signaling pathways and facilitates glycolysis and fatty acid oxidation. Although the results are intriguing, there are some issues that require further attention as detailed below.

Major:

1- Overall, the phenotype of HRD1 LKO mice is increased energy expenditure and lower body weight which –perhaps– explains the lower blood glucose and plasma lipids. However, no insight into how hepatocyte-specific deletion of a gene results in increased energy expenditure is provided. The authors show many interesting findings in liver after HRD1 is deleted (i.e. increased AMPK and AKT activity, etc.) but none of them explains why these mice have increased energy expenditure. It would be nice to speculate a mechanism. For example, is the reason why these LKO mice have increased energy expenditure because they have increased beige/browning which may suggest a liver-adipose tissue crosstalk?

Reply: We agree with the reviewer and now clarified that loss of HRD1 functions leads to systemic alterations to metabolism, which is possibly due to, at least partially, the elevated hepatokine FGF21. This is supported by the following evidences: First we discovered that HRD1 deletion increased the Fgf21 expression from our RNA-Seq data. Second, we confirm the elevation of Fgf21 mRNA by qRT-PCR methods and we also found the FGF21 protein levels in the serum were increased after

hepatic HRD1 deletion. Third, the elevation of energy expenditure of the HRD1 LKO mice was rescued by FGF21 deletion. However, deletion of FGF21 failed to rescue the decreased levels of serum TG and FA in HRD1 LKO mice and Stearoyl-CoA Desaturase (*Scd1*) expression (Fig. 6B) which catalyze the rate-limiting step in the formation of monounsaturated fatty acid, suggesting that HRD1 may regulate TG and FA metabolism through alternative pathways in the absence of FGF21.

Figure S10. Elevation of energy expenditure of HRD1 LKO mice through FGF21 overexpression. (a) Hepatic *Fgf21* mRNA in the WT and L-HRD1 KO mice. (b) Serum FGF21 protein levels in the WT and L-HRD1 KO mice. (c) Energy expenditure of WT, HRD1 LKO and HRD1/FGF21 DKO mice. (d) Serum fatty acid and TG levels of WT, HRD1 LKO and HRD1/FGF21 DKO mice. (e) Relative *Scd1* mRNA expression of WT, HRD1 LKO and HRD1/FGF21 DKO mice.

2- The results presented in Figure 4 and 5 showing that HRD1 deficiency results in activation of AMPK and AKT signaling and lower expression of genes involved in fatty acid and TG synthesis. Are these effects cell intrinsic to hepatocytes? What's the evidence that these changes are NOT due to lower body weight in LKO mice? Similar analysis after acute silencing of HRD1 using primary hepatocytes or hepatocyte cell lines should be done in order to suggest that these results are primarily due to HRD1 deficiency.

Reply: We agree with this reviewer that the altered AMPK and metabolic gene expression in HRD1 LKO mice could be a consequence of reduced body weight instead of a causal factor. To investigate whether the lower expression of genes involved in fatty acid and TG synthesis in the LKO mice are due to lower body weight of the LKO mice, we injected poly I:C into the Mx1-Cre HRD1^{fllox/fllox} mice and their WT littermates to acute silencing of HRD1 in vivo. As showed in Figure 6a, two days after poly-IC treatment when the mice body weight was unaltered, the expression of genes involved in fatty acid synthesis including *Srebf1*, *Chrebf1* and *Scd1* were dramatically decreased in the Mx1-Cre HRD1^{fllox/fllox} mice (Figure 6a), indicating that these changes are not due to lower body weight.

However, we are aware that, as what the reviewer pointed out, even without body weight reduction, the possibility that overactivation of AMPK and lower expression of genes are due to the altered systemic metabolic factors such as serum fat acid levels cannot be fully excluded. We then detected

p-AMPK and p-AKT in the in vitro cultured primary hepatocytes in which the culture media are synchronized. Consistent with our initial observation, AMPK but not AKT activation was significantly increased after HRD1 deletion (Figure 6b). In addition, the expression of genes involved in fatty acid synthesis including *Srebf1*, *Chrebp1*, *Scd1* and *Fads* were also dramatically decreased in HRD1 KO primary hepatocytes (Figure 6c). Collectively, our data indicate that liver HRD1 deletion affect AMPK activation and metabolic response in a hepatocyte-intrinsic manner.

Figure 6a-c. Metabolic alteration induced by hepatic HRD1 deletion was in a hepatocyte-intrinsic manner. (a) Relative mRNA of *Srebf1*, *Chrebp1* and *Scd1* of HRD1^{flox/flox} (WT) and HRD1^{flox/flox} Mx1-Cre⁺ mice 2 days after Ploy I:C injection. (b&c) WT and HRD1 LKO hepatocytes were isolated and cultured overnight. mRNA level of *Srebf1*, *Chrebp1*, *Scd1* and *Fads2* (b) and AMPK and AKT activity were measured (c).

3- Do hepatic HRD1 levels change upon high-fat diet feeding or other models (genetic) of obesity?

Reply: Hepatic HRD1 mRNA was measured after 16 weeks of HFD treatment, no significant changes were found of *Hrd1* mRNA expression.

Figure S1 (a) Relative mRNA levels in a normal chow or after 14 weeks High fat diet treatment.

Minor:

1- Blood glucose is only regulated after refeeding –how about plasma insulin levels? Do HRD1 LKO mice have lower refeed blood glucose levels because they secrete more insulin after feeding?

Reply: Insulin in the serum between WT and LKO mice is no significant difference (Figure S2).

Figure S2 (d) Insulin levels were measured in the serum of WT and HRD1 LKO mice in a refeed condition.

2- Were TG levels in Figure 1e measured in fasting or refeeding? Do TG levels change after refeeding only -like blood glucose?

Reply: TG levels in Figure 1e is measured in a refeed condition and we added the detail information in figure legend of Figure 1e. TG levels were also decreased in a fasted condition as showed below.

3- Although the authors provide H&E staining of liver and show no abnormality, liver enzymes (ALT and AST) should be measured to prove that HRD1 deficiency does not result in liver toxicity.

Reply: We agree with the reviewers that HRD1 deletion may cause liver toxicity. However, analysis of the AST and ALT levels between WT and HRD1 LKO mice did not detect any abnormal increases in the serum of HRD1 LKO mice (Figuer S2).

Figure S2c HRD1 deletion didn't result in liver toxicity. (c) AST and ALT were measured in the serum of WT and HRD1 LKO mice in a refed condition.

4- The degree of hepatic HRD1 silencing should be shown in *Hrd1^{fl/fl} Mx1-Cre* mice.

Reply: We have shown the hepatic HRD1 mRNA levels in Figure S1f.

Figure S1 (f) Hepatic *Hrd1* mRNA 5 days after Poly (I:C) injection.

5- Blood glucose levels in Fig s3a, Fig 2f, Fig 2g are expressed as mM –I believe those are mg/dl.

Reply: This typo is now corrected.

6- Adenovirus-mediated gene delivery is transient and usually lasts 10-14 days. It is interesting to see that adeno-Cre injected mice had less body weight gain even after 1 month of adeno-Cre treatment. It would be nice to provide the liver HRD1 levels to show that the adeno-Cre effect, namely silencing of HRD1, is still present 1 month after adenovirus injection.

Reply: We agree with the reviewer that the adenovirus-mediated gene suppression in mouse liver is often transient. As indicated in the Figure S1f, HRD1 expression was largely inhibited at day 7 but bounced back at 15 days after adenovirus injection, which is similar as reported. Nevertheless, this transient suppression is sufficient to provide a time window for the difference of the body weight accumulation.

7- Figure 2g and 6h are described as insulin tolerance tests in figure legends and written as “glucose infusion rate upon hyperinsulinemic clamping” in the text. The text should be changed to “insulin tolerance test” not to confuse insulin tolerance test with hyperinsulinemic euglycemic clamp studies which is completely different and was not performed in this study.

Reply: We thank the reviewer and the legends of Figure 2 and Figure 6 have been revised as suggested.

8- Figure 2i is missing significance asterisk/signs.

Reply: This is now corrected.

9- Figure 4c blots are too dark and hard to interpret. It would be nice to see lower exposed western blot data to clearly argue that ubiquitination is decreased.

Reply: The samples were blotted with anti-Ub again and exposed shorter. As showed in Figure 4c, the endogenous ubiquitination level of HSD14B4 was decreased after hepatic HRD1 deletion.

10- Figure 4d is missing the actin blots. The densitometric quantification is done with actin and not shown in the figures.

Reply: The GAPDH loading control is now added.

Reviewer #3 (Remarks to the Author):

Identifying environmental and genetic factors that disrupt metabolic processes in obesity and chronic over-nutrition are a critical avenue of scientific inquiry for addressing metabolic disease clusters in modern society. Despite much progress, basic features driving pathogenesis remain unclear. Here, Wei & Yuan et al consider the broad contribution of dysregulated catalytic enzyme activity in the collective concept of metabolic syndrome, focusing on the relatively unexplored area of protein degradation/turnover. This inquiry leads the authors to a detailed investigation of HRD1, an E3 ligase known to play a role in targeting degradation of misfolded proteins in the ER by the process of ERAD. Based on this foreknowledge, one might have expected genetic disruption of HRD1 in livers of high fat diet fed mice, which are known to exhibit elevated ER stress, would have caused severe problems in liver metabolism. Quite the contrary however, Wei & Yuan et al show that these animals display several phenotypic improvements including higher energy expenditure, reduced weight gain & adiposity, and protection against hepatic steatosis as well as better lipid profiles, insulin sensitivity, and glucose homeostasis.

The authors further show using unbiased proteomic analysis that the phenotype observed in the HRD1 deficient mice seems to be less related to ERAD but instead through a previously unrecognized role of HRD1 in regulating turnover rates of key metabolic enzymes ENTPD5, CPT2, RMND1, and HSD17B4, as well as some other expected targets. Overall, this study has several novel and powerful elements compelling enough to support its publication. That being said, I have a few issues that I think are worth consideration.

Issues:

1) The overall study implies that nutrients regulate HRD1 in the liver following feeding, causing HRD1 to cause degradation of these enzymes. As such, deletion of HRD1 causes the enzymes to be expressed much higher. However, the authors really only look at the role of fasting/feeding on HRD1 levels (and not its actual activity) in one panel (Figure 1b). I think this should be expounded upon. Specifically, it would be highly desirable that the authors correlate the feeding induced rise in HRD1 levels with the levels of at least one of the enzymes mentioned (ENTPD5, CPT2, RMND1 or HSD17B4). Whether the activity responds to feeding and regulation is lost in HRD1 deficiency is of interest and would add to the conclusions.

2) Along these lines, the authors state “we found that HRD1 was postprandially induced to control the metabolic balance in the physiological condition.” However, the authors do not discuss the role of such a regulatory circuit in a physiological setting. Specifically, how might and for what function would HRD1 target these enzymes following a meal?

Reply: We totally agree with the reviewer that, since refeeding dramatically induces HRD1 expression in liver, it would be interesting to determine whether this induction of HRD1 expression leads to destruction of its target proteins. As shown in Figure S10, a significant reduction in the protein expression of both ENTPD5 and CPT2 was detected in the liver tissue from refeed mice. This reduction is likely due to the ubiquitination catalyzed by elevated HRD1 because the ubiquitination of ENTPD5 and CPT2 in the liver of refeed mice was increased. However, we also noticed that while HRD1 deletion results in elevated HSD17B4 and RMND1 protein expression, a 4-hour refeeding did not lead to a dramatic reduction of their protein expressions. Several possibilities: for example, these HSD17B4 and RMND1 may have a longer half-life or with a different dynamic in ubiquitination-mediated degradation.

Therefore, our results imply a possible molecular mechanism by which refeeding induced HRD1 expression in regulating liver metabolism through targeting both ENTPD5 and CPT2 for degradation. Interestingly, analysis of the gene expression profiles in the liver WT and HRD1 LKO mice indicate that a substantial number of genes was dramatically increased in HRD1-null liver only in the refeed condition, and much less genes were altered in fasted condition (Figure S10d&e, 1249 vs 237). GO function analysis indicates that these genes are critical for multiple liver metabolic response pathways including steroid hormone mediated signaling pathway, glucose and lipid metabolic process and circadian rhythm (Figure S10f). This has been now discussed in the revised manuscript.

Figure 6g-j & Figure S12. HRD1 was postprandially induced to control the metabolic balance in the physiological condition. (a) Hepatic ENTPD5, CPT2, RMND1 and HSD17B4 protein levels were measured in the fasted and refeed conditions **(b&c)** Ubiquitination levels of ENTPD5 and CPT2 were

measured in the fasted and refeed conditions. (d) Differential genes in the fasted and refeed genes were overlapped. (e) Volcano plot of the differential genes from only in the refeed condition. (f) Metabolic functions of the genes from e.

3) While the phenotype of the HRD1 deficient model is striking, the energy expenditure result is almost bewildering. Can the authors explore how a change in liver metabolism has such a profound effect on whole body energy expenditure? Any insight into this would be very interesting and at the very least the discussion should be expanded.

Reply: We agree with the reviewer and now clarified that loss of HRD1 functions leads to systemic alterations to metabolism, which is possibly due to, at least partially, the elevated hepatokine FGF21. Chronic exposure to FGF21 markedly induces the growth retardation, female infertility and energy expenditure elevation in mice. This is supported by the following evidences: First we discovered that HRD1 deletion increased the Fgf21 expression from our RNA-Seq data. Second, we confirm the elevation of Fgf21 mRNA by qRT-PCR methods and we also found the FGF21 protein levels in the serum were increased after hepatic HRD1 deletion. Third, the elevation of energy expenditure of the HRD1 LKO mice was rescued by FGF21 deletion. However, deletion of FGF21 failed to rescue the decreased levels of serum TG and FA in HRD1 LKO mice and Stearoyl-CoA Desaturase (Scd1) expression (Fig. 6B) which catalyze the rate-limiting step in the formation of monounsaturated fatty acid, suggesting that HRD1 may regulate TG and FA metabolism through alternative pathways in the absence of FGF21.

Figure S10. Elevation of energy expenditure of HRD1 LKO mice through FGF21 overexpression. (a) Hepatic Fgf21 mRNA in the WT and L-HRD1 KO mice. (b) Serum FGF21 protein levels in the WT and L-HRD1 KO mice. (c) Energy expenditure of WT, HRD1 LKO and HRD1/FGF21 DKO mice. (d) Serum fatty acid and TG levels of WT, HRD1 LKO and HRD1/FGF21 DKO mice. (e) Relative Scd1 mRNA expression of WT, HRD1 LKO and HRD1/FGF21 DKO mice.

4) On line 160 and 161 of the text it describes the use of hyperinsulinemic clamps in figure 2f. However, figure 2f shows a glucose tolerance test, not a clamp study. Please correct.

Reply: Thanks for this comment; we changed our words to “WT mice on the HFD showed impaired glucose tolerance and insulin sensitivity compared to HFD-fed HRD1 LKO mice In Figure 2 and

Figure 6.

5) Detailed information should be provided in the generation and housing of the control and HRD1-deficient mice, the nature of the diets, and their genetic background.

Reply: We described our detail information of the generation and housing of mice as follow:

Hrd1-targeting vector was generated as in Fig. S1a, and then transfected into an embryonic stem cell line generated from C57BL/6 mice. Neomycin selects were screened by PCR. Seven clones were obtained and confirmed by Southern blotting. Blastocyst injections resulted in several chimeric mice with the capacity for germline transmission. Breeding of heterozygous mice yielded Hrd1^{flox/flox} mice without phenotypic abnormalities in expected Mendelian ratios. Mx1-Cre and Albumin-Cre mice are at the C57BL/6 genetic background and purchased from The Jackson Laboratory. All mice were bred in a specific pathogen-free facility, and all animal experiments were approved by the institutional animal care and use committees (IACUC) at Northwestern University. Animals were maintained on a standard chow diet under 12-hour light and dark cycles beginning at 5:00 a.m. and 5:00 p.m., respectively. The High fat diet (45 kcal% fat) is from the Research Diet Inc.

Reviewer #4 (Remarks to the Author):

The manuscript by Wei et al. describes the detailed analysis of HRD1 function in the liver of mice. While the protein was described as a regulator of ER stress, the authors here show a link of this ubiquitin ligase with various aspects of metabolism. They reveal an important role for the protein in energy expenditure and demonstrate that loss of the protein results in resistance against high-fat diet-induced obesity and liver steatosis. A conditional mouse model supports the targeting of this protein as a therapeutic protein.

Experiments are well performed and overall the data looks convincing and of high quality. The manuscript is well written and has a logical structure.

I only have a few minor comments on the manuscript:

I think that the current title of the manuscript ('Hepatic ER-associated ubiquitin ligase HRD1 is a therapeutic target for high-fat diet-induced metabolic syndrome') should be less focused on the therapeutic aspect of the HRD1 protein. While there is some support that HRD1 can be a good therapeutic target, there is still some additional data needed to confirm this. Moreover, the authors do not really discuss how such a therapeutic targeting of HRD1 would be realized.

For the proteomics experiments it would be good if the authors clearly describe how many controls were actually performed. From the SC numbers I am assuming that 2 controls were performed. More controls would have been better. Since SC numbers are markedly different and since the authors provide sufficient validation for the selected candidates the data is overall very convincing.

Reply: Two controls were performed for the IP-MS experiment of HRD1, which was described in the modified manuscript (Page 22).

Combined use of COMPASS and SAINT is a solid strategy to prioritize candidates. It is not fully clear what drove the selection of the candidates for further validation. Were additional criteria used or were these proteins used to represent certain biological processes?

Reply: As the lipid accumulations in the liver and adipose tissue were decreased in HRD1 KO mice, we proposed that HRD1 deletion may inhibit the body weight gain through inhibiting the lipogenesis and/or increasing the fatty acid oxidation. As published before, ENTPD5, together with CMPK1 and AK1, constitute an ATP hydrolysis cycle that converts ATP to AMP, resulting in AMPK pathway overactivation and lipogenesis down-regulation. CPT2, RMND1, HDS17B4, ATP5D, MLYCD and AKR1C1 are enzymes which are directly involved in fatty acid metabolism.

Did the authors use the CRAPome for filtering?

Due to that no contaminant data was available for HepG2 cells, we did not use the CRAPome for filtering the IP-MS data. The common contaminants were filtered by cRAP proteins sequences from gpmDB (<http://www.thegpm.org/crap/index.html>) and contaminants from MaxQuant (<http://www.coxdocs.org/doku.php?id=maxquant:start>). The related description was added in the **Methods and Materials** of the modified manuscript.

The proteomics data should be uploaded to a repository (e.g. PRIDE) so it is available for other scientists.

Reply: Thanks for the comments. The raw proteomic data was deposited in a public repository, iProX (<http://www.iprox.org/index>), with identifier IPX0000928000. All of the raw data can be downloaded from <http://www.iprox.org/page/PSV023.html?url=1493008523383O9Vv> with password (o8IL). The raw data will be public accessible after publication of the manuscript. The corresponding description was modified in **Methods and Materials** of the revised manuscript.

RT-qPCR experiments are more reliable when more housekeeping genes are used.

Reply: All RT-qPCR experiments were performed with *Tuba* as a housekeeping gene. This is now clarified.

Fig 3g. Please specify clearly if the detection of the candidate interactors is with an epitope tag (overexpression) or with specific antibodies.

Reply: In Figure 3g show the interaction of the endogenous HRD1 with its partners, but not in transiently transfected cells. This is now clarified.

Full gel views should be made available in supplementary information.

Reply: All images for the full gels are available and will be provided by the acceptance based on the Journal policy.

Some typos and other suggested improvements:

Abstract line 22: please start with the full name of the protein

Line 29: Genome-wide mRNA sequencing

Line 105: Similarly, we detected dramatically lower...

Fig S6: resolution currently not sufficient. Integration of the provided figure is required.

Reply: All have been corrected in the revised manuscript.

Reviewers' comments:

Reviewer #1 (Remarks to the Author):

In this revised manuscript, the authors have added quite a bit of experimental evidence to support their conclusions.

My opinion of the original manuscript was that the data were already fairly robust, but that the authors needed to temper some of their conclusions to be more cautious.

In the revision, the authors addressed some of these critiques experimentally, rendering some of the original comments moot. However, there are still a couple of points that I think need to be addressed by textual revision so that the manuscript better reflects the data:

1. The added experiment with a PERK inhibitor is very difficult to interpret. As the changes in expression of the relevant genes are suppressive rather than stimulatory, the effect of a very short (3h) treatment is going to depend highly on the turnover of these mRNAs. I don't dispute that the metabolic effects being seen in the KOs might be independent of ER stress, but I don't think the authors can rule the ER stress possibility out. Their writing should allow for that possibility, even though it is not their preferred hypothesis

2. My original point about not being able to come to conclusions about the activity of various metabolic pathways still stands. As the authors concede, they do not have the technical ability to assay these pathways directly. Thus, they need to be more conservative in their wording. Especially in the abstract, which currently states: "Genome-wide mRNA sequencing revealed that HRD1-deficiency reprograms liver metabolic gene expression profiles, suppressing both glycogenesis and lipogenesis but facilitating glycolysis and fatty acid oxidation." A more accurate sentence would be something like: "Genome-wide mRNA sequencing revealed that HRD1-deficiency reprograms liver metabolic gene expression profiles, including suppressing genes involved in glycogenesis and lipogenesis but upregulating genes involved in glycolysis and fatty acid oxidation." Similar caution should be exercised in the main text.

I want to be clear that I am not not asking for new experiments (nor was I in the original critique)-just more cautious interpretation and writing.

Reviewer #2 (Remarks to the Author):

The revised paper submitted by the authors responded satisfactory to most of the comments raised previously. However, I would like to point out a few more details for the authors to check:

1- In the Introduction section, authors argue that "Genetic HRD1 suppression specifically in liver alone was sufficient to protect mice from HFD-induced obesity, hyperlipidemia, hyperglycemia, NAFLD, and insulin resistance. This suggested that HRD1 is the main downstream effector of ER stress to increase glucose and lipid levels. ---". However, their results using PERK inhibitor clearly suggests that HRD1 regulates lipid metabolism in an ER stress (or PERK pathway to be clear) - independent way (hence their response to the first reviewer). I suggest them to remove this conclusion from the introduction or make it clear it that "ER stress was not the reason" for the metabolic effects that they see in the HRD1 deficient mice.

2- In the discussion section, it is written as "HRD1/FGF21 will be generated to further characterize the molecular mechanism of the increased energy expenditure induced by hepatic HRD1 deletion.". Since the authors have done that experiment (Figure S10), I suggest discussing those findings here instead.

3- The authors argue that activation of IRE1a has been shown to impair insulin signaling through JNK and Xbp1-PI3K interaction and cite Park et al 2010 for Xbp1-PI3K. However that study specifically shows that p85 (PI3K) interacts with, and increase the nuclear translocation of spliced XBP-1 –not Xbp1 impairing insulin signaling or PI3K activation. I suggest removing that citation.

4- I was unable to see the text for Figure S2d results –please insert in the text or remove the figure.

5- Figure 2K legend should be “energy expenditure” not expansion.

6- Figure 7h is missing asterisk to show significant difference.

7- There are a few typos on lines: 99, 145, and 333

8- Fgf21 might be the key driver and expanding the characterization of the double Hrd1/Fgf21 KO mice in the original manuscript would be beneficial. It looks like fgf21 is responsible for many effects of L-Hrd1 KO mice like rescuing the growth retardation and circadian regulation.

9- I also think that the “growth retardation” characterized in the second manuscript is troublesome and might be responsible for the phenotype described on the original manuscript. The findings described in the original manuscript could be secondarily due to the “growth retardation effect”.

Reviewer #3 (Remarks to the Author):

The authors have provided a robust response to the questions raised during the initial review and I have no further comments.

Reviewer #4 (Remarks to the Author):

The comments that I have made to the manuscript by Wei et al. have been sufficiently addressed. The manuscript improved significantly by addressing the comments from the reviewers. The mechanistic insights provided by the authors also strengthen the story. The implication of the FGF21 axis (and the apparent overall importance of HRD1 in liver) raises questions on the potential of targeting HRD1 with small molecules without off-target effects. Despite the changed title, therapeutic targeting is still an important part of the message of the manuscript (in abstract and start of the discussion). As pointed out by the authors, the availability of a small molecule is promising. I therefore feel that the claimed therapeutic potential should be framed better in the discussion. The small molecule for HRD1 was also developed for treatment in the liver. How does this relate to the story presented here? How realistic is the use for the conditions that are discussed in the manuscript?

Minor:

The full name of HRD1 should be in the abstract as well.

It is unclear what the authors meant by their comment on the question around the use of housekeeping genes for RT-qPCR. The profound effects of HRD1 deletion may also affect some of the housekeeping genes. It is therefore a good idea to measure and combine different housekeeping genes to ensure a good reference.

We appreciate the efforts for you and all the four reviewers in reviewing our manuscript and their recognition of the significance of our study. All the concerns have been carefully addressed in a point-to-point manner as indicated in this rebuttal letter as well as in the revised manuscript.

Reviewer #1 (Remarks to the Author):

In this revised manuscript, the authors have added quite a bit of experimental evidence to support their conclusions.

My opinion of the original manuscript was that the data were already fairly robust, but that the authors needed to temper some of their conclusions to be more cautious.

In the revision, the authors addressed some of these critiques experimentally, rendering some of the original comments moot. However, there are still a couple of points that I think need to be addressed **by textual revision** so that the manuscript better reflects the data:

1. The added experiment with a PERK inhibitor is very difficult to interpret. As the changes in expression of the relevant genes are suppressive rather than stimulatory, the effect of a very short (3h) treatment is going to depend highly on the turnover of these mRNAs. I don't dispute that the metabolic effects being seen in the KOs might be independent of ER stress, but I don't think the authors can rule the ER stress possibility out. Their writing should allow for that possibility, even though it is not their preferred hypothesis

Reply: We have now clarified our conclusion as suggested that our results from the experiment with a short time of PERK inhibitor treatment, while further support our initial conclusion, the possibility that ER stress is involved in liver metabolic changes by Hrd1 deletion cannot be fully excluded. This has been discussed in the revised manuscript.

2. My original point about not being able to come to conclusions about the activity of various metabolic pathways still stands. As the authors concede, they do not have the technical ability to assay these pathways directly. Thus, they need to be more conservative in their wording. Especially in the abstract, which currently states: "Genome-wide mRNA sequencing revealed that HRD1-deficiency reprograms liver metabolic gene expression profiles, suppressing both glycogenesis and lipogenesis but facilitating glycolysis and fatty acid oxidation." A more accurate sentence would be something like: "Genome-wide mRNA sequencing revealed that HRD1-deficiency reprograms liver metabolic gene expression profiles, including suppressing genes involved in glycogenesis and lipogenesis but upregulating genes involved in glycolysis and fatty acid oxidation." Similar caution should be exercised in the main text.

Reply: We agree with this overstatement, both the abstract and the main text have been revised as suggested.

I want to be clear that I am not not asking for new experiments (nor was I in the original critique)--just more cautious interpretation and writing.

Reviewer #2 (Remarks to the Author):

The revised paper submitted by the authors responded satisfactory to most of the comments raised previously. However, I would like to point out a few more details for the authors to check:

1- In the Introduction section, authors argue that "Genetic HRD1 suppression specifically in liver alone was sufficient to protect mice from HFD-induced obesity, hyperlipidemia, hyperglycemia, NAFLD, and insulin resistance. This suggested that HRD1 is the main downstream effector of ER stress to increase glucose and lipid levels. ---". However, their results using PERK inhibitor clearly suggests that HRD1 regulates lipid metabolism in an ER stress (or PERK pathway to be clear) -independent way (hence their response to the first reviewer). I suggest them to remove this conclusion from the introduction or make it clear it that "ER stress was not the reason" for the metabolic effects that they see in the HRD1 deficient mice.

Reply: This conclusion is now removed from the revised manuscript and clarified as suggested by both Reviewer # 1 (above) and 2.

2- In the discussion section, it is written as "HRD1/FGF21 will be generated to further characterize the molecular mechanism of the increased energy expenditure induced by hepatic HRD1 deletion.". Since the authors have done that experiment (Figure S10), I suggest discussing those findings here instead.

Reply: A discussion is added in the revised manuscript as suggested.

3- The authors argue that activation of IRE1a has been shown to impair insulin signaling through JNK and Xbp1-PI3K interaction and cite Park et al 2010 for Xbp1-PI3K. However that study specifically shows that p85 (PI3K) interacts with, and increase the nuclear translocation of spliced XBP-1 –not Xbp1 impairing insulin signaling or PI3K activation. I suggest removing that citation.

Reply: This citation has been removed.

4- I was unable to see the text for Figure S2d results –please insert in the text or remove the figure.

Reply: The citation of this figure s2d is added to the main text.

5- Figure 2K legend should be "energy expenditure" not expansion.

6- Figure 7h is missing asterisk to show significant difference.

7- There are a few typos on lines: 99, 145, and 333

Reply: These typos have been corrected.

8- Fgf21 might be the key driver and expanding the characterization of the double Hrd1/Fgf21 KO mice in the original manuscript would be beneficial. It looks like fgf21 is responsible for many effects of L-Hrd1 KO mice like rescuing the growth retardation and circadian regulation.

9- I also think that the "growth retardation" characterized in the second manuscript is troublesome and might be responsible for the phenotype described on the original manuscript. The findings described in the original manuscript could be secondarily due to the "growth retardation effect".

Reply Q8&9: While the increased Fgf21 is a critical mechanism leading to significant increases in energy expenditure, growth retardation and circadian dysregulation in Hrd1 LKO mice as presented in another manuscript, we conclude in the current manuscript that "ER-associated ubiquitin ligase HRD1 programs liver metabolism through targeting multiple metabolic enzymes in particular the Entpd5-AMPK pathway and is largely independent on Fgf21 because:

(1) Hrd1 deletion protects mice from fatty liver disease is unlikely a consequence of growth retardation, because the inducible Hrd1 deletion in the adult mice when the body weight and sizes were fully

developed at 12 weeks of age and indistinguishable between WT and *Hrd1^{f/f}*Mx-Cre mice still sufficiently protected them from HFD-induced fatty liver disease (Fig. 7).

- (2) Hepatic HRD1 suppression leads to a significant reduction in the serum levels of free fatty acid and TG in a FGF21-independent manner because further *Fgf21* deletion did not restore the levels of free fatty acid and TG (Figure S10d);
- (3) The decrease in the expression of the lipogenesis genes and increased of the fatty acid oxidation genes are rescued by *Entpd5* knockdown and AMPK inhibitor (Figure 6d & f), but not *Fgf21* deletion (Figure S10e), clearly indicating that HRD1 control the lipogenesis and fatty acid oxidation through the *Entpd5*-AMPK but not FGF21 pathway.
- (4) To further clarify, we have performed new experiments by analyzing the liver tissues in HFD-feeding WT, *Fgf21* ko, *Hrd1* KO and *Hrd1/Fgf21* double KO mice. Consistent with our original conclusion, genetic HRD1 suppression TOTALY protected mice from HFD-induced lipid accumulation EVEN when *Fgf21* gene is deleted, clearly indicating that hepatic HRD1 inhibition protects mice from HFD-induced fatty liver disease independent of *Fgf21* (Figure s11 and below).

Figure S11. Reduction of lipid accumulation in HRD1 LKO mice was not through FGF21 overexpression. H&E (top panels) and Oil Red O (bottom panels) staining of the WT, *Fgf21^{-/-}*, *Hrd1^{-/-}* and *Fgf21^{-/-}Hrd1^{-/-}* livers after 14 weeks HFD treatment.

Therefore, our data collectively conclude that ER-associated ubiquitin ligase HRD1 programs liver metabolism through targeting multiple metabolic enzymes in particular the *Entpd5*-AMPK pathway and largely, if not totally, in an *Fgf21*-independent manner.

Reviewer #3 (Remarks to the Author):

The authors have provided a robust response to the questions raised during the initial review and I have no further comments.

Reviewer #4 (Remarks to the Author):

The comments that I have made to the manuscript by Wei et al. have been sufficiently addressed. The manuscript improved significantly by addressing the comments from the reviewers. The mechanistic insights provided by the authors also strengthen the story. The implication of the FGF21 axis (and the apparent overall importance of HRD1 in liver) raises questions on the potential of targeting HRD1 with small molecules without off-target effects. Despite the changed title, therapeutic targeting is still an important part of the message of the manuscript (in abstract and start of the discussion). As pointed out by the authors, the availability of a small molecule is promising. I therefore feel that the claimed therapeutic potential should be framed better in the discussion. The small molecule for HRD1 was also developed for treatment in the liver. How does this relate to the story presented here? How realistic is the use for the conditions that are discussed in the manuscript?

Reply: A more comprehensive discussion of the potential use of Hrd1-specific inhibitors in treatment of fatty liver disease is added to the revised manuscript as suggested.

Minor:

The full name of HRD1 should be in the abstract as well.

It is unclear what the authors meant by their comment on the question around the use of housekeeping genes for RT-qPCR. The profound effects of HRD1 deletion may also affect some of the housekeeping genes. It is therefore a good idea to measure and combine different housekeeping genes to ensure a good reference.

Reply: We actually have used two housekeeping genes (Tubulin and b-actin) for the analysis and similar results were obtained.

REVIEWERS' COMMENTS:

Reviewer #2 (Remarks to the Author):

The authors have responded to the questions and edited the manuscript according to reviewers' suggestions. I have no further comments.